# TEST-TIME ADAPTATION FOR CROSS-MODAL RETRIEVAL WITH QUERY SHIFT

**Haobin Li**[1]  **Peng Hu**[1]  **Qianjun Zhang**[2]  **Xi Peng**[1,3]  **XitingLiu**[4]  **Mouxing Yang**[1*]
College of Computer Science, Sichuan University, China.[1]  Southwest Jiaotong University, China.[2]
State Key Laboratory of Hydraulics and Mountain River Engineering, Sichuan University, China.[3]
Georgia Institute of Technology, USA.[4]
{haobinli.gm, penghu.ml, pengx.gm, liu.xt0617, yangmouxing}@gmail.com, zqjblue@163.com

## ABSTRACT

The success of most existing cross-modal retrieval methods heavily relies on the assumption that the given queries follow the same distribution of the source domain. However, such an assumption is easily violated in real-world scenarios due to the complexity and diversity of queries, thus leading to the query shift problem. Specifically, query shift refers to the online query stream originating from the domain that follows a different distribution with the source one. In this paper, we observe that query shift would not only diminish the uniformity (namely, within-modality scatter) of the query modality but also amplify the gap between query and gallery modalities. Based on the observations, we propose a novel method dubbed Test-time adaptation for Cross-modal Retrieval (TCR). In brief, TCR employs a novel module to refine the query predictions (namely, retrieval results of the query) and a joint objective to prevent query shift from disturbing the common space, thus achieving online adaptation for the cross-modal retrieval models with query shift. Expensive experiments demonstrate the effectiveness of the proposed TCR against query shift. Code is available at https://github.com/XLearning-SCU/2025-ICLR-TCR.

## 1 INTRODUCTION

Given queries of interest, cross-modal retrieval (Lee et al., 2018; Yang et al., 2022b; Yan et al., 2023; Lin et al., 2024; Ma et al., 2024a) try to associate some relevant samples from the gallery set across various modalities, supporting numerous applications such as intelligent surveillance and search engine. The key to cross-modal retrieval is learning a well-established common space, hoping to distinguish different instances within the same modality while gathering the same instance across different modalities. Recently, the pre-trained models (Jia et al., 2021; Yang et al., 2022a; Li et al., 2023b; Huang et al., 2024) have emerged as the dominant paradigm for cross-modal retrieval. As shown in Fig. 1(a), after acquiring generic knowledge from the source domain, the pre-trained models can either perform zero-shot retrieval in the target domains or be fine-tuned on domain-specific data for customization.

Despite the promising performance of the pre-trained models, their success heavily relies on the assumption that the given queries exactly follow the same distribution from the source domain, which is hard to satisfy in real-world applications. Specifically, as shown in Fig. 1(b), inquirers might embrace different cultural backgrounds or enjoy their individual preferences, resulting in the online query stream derived from either scarce or highly personalized domains. Clearly, such out-of-domain queries violate the identical distribution assumption and thus lead to the *query shift* problem. As a result, the existing cross-modal retrieval models fail to handle the query shift and inevitably suffer from significant performance degradation, leaving an urgent need to develop an online adaptation method for addressing the query shift problem.

As one of the most effective paradigms in reconciling distribution shifts, Test-Time Adaptation (TTA) methods (Wang et al., 2021; Press et al., 2023; Bar et al., 2025; Dong et al., 2025) work by continually updating the given source model using the online target data stream. Although achieving

---

*Corresponding author.

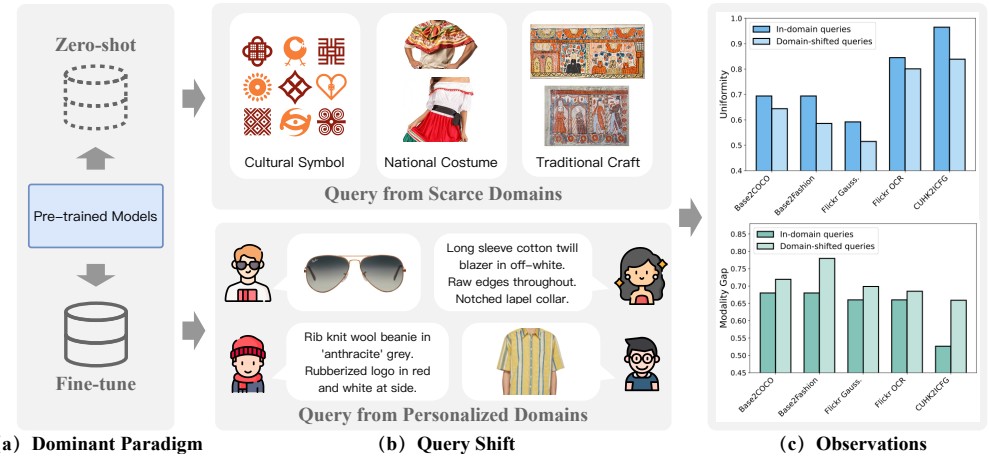

**(a) Dominant Paradigm**        **(b) Query Shift**        **(c) Observations**

Figure 1: (a) **Dominant Paradigm**: the pre-trained models embrace powerful zero-shot retrieval capacity and could be fine-tuned on domain-specific data for customization, which has emerged as the dominant paradigm for cross-modal retrieval. (b) **Query Shift**: the performance of the paradigm would be significantly degraded when encountering the query shift problem. On the one hand, collecting sufficient data to tailor the pre-trained models for scarce domains is daunting and even impossible. On the other hand, as the saying goes, "Different strokes for different folks", even fine-tuned models cannot accommodate all personalized domains. (c) **Observations**: we study the query shift problem for cross-modal retrieval and reveal the following observations. Namely, query shift not only diminishes the uniformity of the query modality but also amplifies the modality gap between the query and gallery modalities, undermining the well-structured common space inherited from pre-trained models.

great success, it is intractable to adopt existing TTA methods for cross-modal retrieval with query shift due to the following two reasons. On the one hand, most existing TTA methods focus on the unimodal setting while overlooking the complexity of the query shift in the cross-modal setting. More specifically, the query shift in the cross-modal setting would not only affect the intra-modality distribution but also hinder the cross-modal alignment. On the other hand, most existing TTA methods are specifically designed for the recognition task, which would struggle with the heavy noise from the query predictions if simply applied to the retrieval task. Intuitively, for a given sample or query, it has a random probability of $1/K$ or $1/N$ to be correctly associated with the desirable category or cross-modal counterpart in the recognition task or retrieval task, where $K$ and $N$ denote the class number and candidate number, respectively, with $N \gg K$.

To specifically develop a TTA method for cross-modal retrieval with the query shift, we first present two key observations as illustrated in Fig. 1(c). To summarize, we conclude that the query shift would diminish the uniformity of the query modality, prohibiting discrimination between diverse queries in the common space. Moreover, query shift would amplify the modality gap between query and gallery modalities, undermining the well-constructed common space established by the pre-trained models.

Based on the above observations, we propose achieving robust cross-modal retrieval against the query shift by endowing the existing TTA methods with the capacity to manipulate both the modality uniformity and modality gap. To be specific, we propose a novel method, dubbed Test-time adaptation for Cross-modal Retrieval (TCR), which consists of a novel query prediction refinement module and a novel joint objective function. First, the query prediction refinement module is adopted to refine the retrieval results of the existing TTA methods and thus obtain the retrieval-favorable predictions for queries. After that, the joint objective is employed on the refined query predictions to achieve online adaptation for cross-modal retrieval models under query shift. More specifically, the joint objective function is composed of three individual losses that embrace the following merits. To enhance the uniformity of the query modality, the intra-modality uniformity learning loss performs contrast between queries and their respective centers, thus guaranteeing the discrimination between queries. To rectify the modality gap between the query and gallery modalities, the inter-modality gap learning loss narrows the difference between the query and gallery modalities with the plausible constraint estimated from off-the-shelf models, thus inheriting the well-established common

space. To prevent overfitting on noisy query predictions, the noise-robust adaptation loss amplifies the contribution of high-confident predictions while alleviating the noisy ones with a self-adaptive threshold. The major contributions of this work could be summarized as follows.

- To the best of our knowledge, this work could be one of the first studies on the query shift problem, revealing its underlying impacts on cross-modal retrieval. Specifically, the query shift would not only diminish the uniformity of the query modality but also amplify the modality gap between query and gallery modalities, undermining the well-established common space derived from the source model.

- We propose a novel test-time adaptation method named TCR. TCR first employs a novel module to refine the query predictions, thus supporting the existing TTA methods for cross-modal retrieval. Then, TCR adopts a novel objective function that can not only manipulate both the modality uniformity and modality gap but also prevent the model from overfitting noisy query predictions, thus achieving robust cross-modal retrieval with query shift.

- Extensive experiments verify the effectiveness of the proposed method. Furthermore, we benchmark the existing TTA methods on cross-modal retrieval with query shift across six widely-used image-text datasets, hoping to facilitate the study of test-time adaptation beyond unimodal tasks.

## 2 RELATED WORK

In this section, we briefly review two topics related to this work, *i.e.*, domain adaptation for cross-modal retrieval and test-time adaptation.

### 2.1 DOMAIN ADAPTATION FOR CROSS-MODAL RETRIEVAL

Cross-modal retrieval aims to establish a well-structured common space, where semantically-relevant candidates could be prioritized for the queries. However, most existing cross-modal retrieval methods implicitly assume that the given queries follow the same distribution as the source data. Unfortunately, such an ideal assumption is easily violated due to the complexity of real-world applications, leading to the query shift problem as discussed in Introduction. To address the problem, some Unsupervised Domain Adaptation (UDA) methods have been proposed to reconcile the distribution differences for robust cross-modal retrieval. Based on the way to achieve robustness against query shift, these approaches could be roughly grouped into the following three categories: i) pseudo-labeling methods (Munro et al., 2021; Hao et al., 2023), which first select the most relevant cross-modal pairs as positives while treating the irrelevant pairs as negatives and then conduct metric learning upon the pairs to adapt the model for target domains; ii) domain alignment methods (Peng & Chi, 2019), which mitigate distribution discrepancies between the target and source domains by resorting to the maximum mean discrepancy minimization or mutual information minimization approaches; iii) prototype-based methods (Liu et al., 2021a; Chen et al., 2021), which first constructs different sets of prototypes to represent various domains and then achieves domain adaptation by minimizing the KL divergence between the corresponding prototype sets.

Despite the promising performance, the existing domain adaptation works for cross-modal retrieval require accessing the entire target domain. As a result, these works cannot achieve adaptation for the online query stream, limiting their practicability in real-time scenarios such as search engine. Different from them, this paper proposes a new adaptation method for addressing the query shift problem, which could be one of the first online adaptation approaches for cross-modal retrieval.

### 2.2 TEST-TIME ADAPTATION

Test-time Adaptation (TTA) has emerged as a promising avenue for domain adaptation, which aims to reconcile the distribution shifts in an online manner. Towards achieving this goal, some Test-Time Training (TTT) approaches (Sun et al., 2020; Liu et al., 2021b; Gandelsman et al., 2022) have been proposed, which require modifying the training process of the source model and adding an auxiliary self-supervised task. As a result, the source model could be adapted by performing the self-supervised task on the online target data stream. To avoid the reduplicated training cost of the source model, Fully Test-Time Adaptation (Wang et al., 2021) paradigm has been proposed, which could

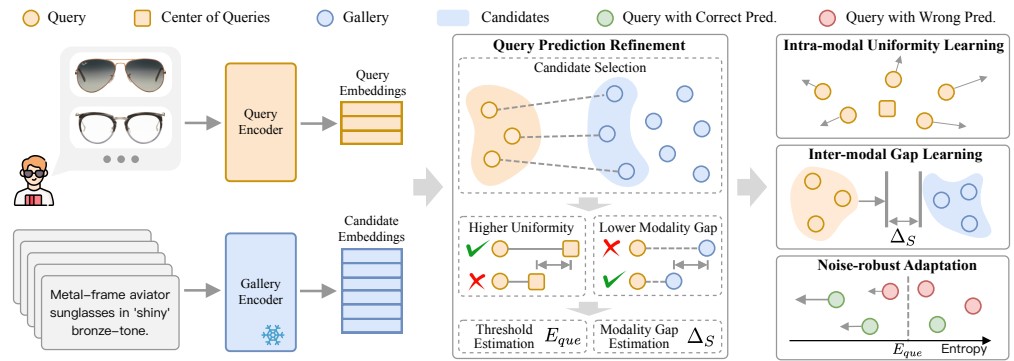

Figure 2: Overview of the proposed TCR. For the given online queries, the modality-specific encoders are employed to project the query and gallery samples into the latent space established by the source model. The obtained query embeddings and gallery embeddings are passed into the query prediction refinement module. In the module, TCR first selects the most similar gallery sample for each query and obtain the query-gallery pairs. After that, the pairs with higher uniformity and lower modality gap are chosen to estimate the filtering threshold of query predictions and modality gap of the source model as the constraints for the adaptation. Finally, three loss functions are employed to achieve robust adaptation for cross-modal retrieval with query shift.

be coarsely divided into the following three categories: i) online TTA methods (Chen et al., 2022; Lee et al., 2024), which continually update the normalization layers by resorting to the unsupervised objectives, such as entropy minimization or its variants. ii) robust TTA methods (Niu et al., 2022; 2023; Zancato et al., 2023), which strive to improve the robustness against noisy predictions, mixed distribution shifts, label shifts, and so on. iii) TTA beyond recognition, which focuses on the tasks including but not limited to image restoration (Gou et al., 2024), multimodal recognition (Yang et al., 2024), and multimodal segmentation (Cao et al., 2023).

In this paper, we focus on online TTA for achieving robust cross-modal retrieval against query shift. Among existing approaches, DISC (Ma et al., 2024b) is most relevant to our work, while having significantly different motivations. In brief, DISC is designed to adapt the pre-trained image hashing models against the distribution shift for achieving effective retrieval in the target domain. Different from DISC, our work focuses on addressing the query shift problem for cross-modal retrieval, which is less-touched by the existing studies. Moreover, unlike DISC that requires accessing the source data to train the hashing model, our TCR could adapt the off-the-shelf pre-trained models without using the source data.

## 3 METHOD

In this section, we introduce the proposed Test-time adaptation for Cross-modal Retrieval (TCR) to handle the query shift problem for cross-modal retrieval. The section is structured as follows. In Section 3.1, we present the formal definition of the query shift problem and design a simple baseline to facilitate TTA for cross-modal retrieval. In Section 3.2, we propose the query prediction refinement module to derive the retrieval-favorable query predictions. In Section 3.3-3.5, we design a novel joint objective function to achieve robust cross-modal retrieval against query shift.

### 3.1 NOTATIONS AND PROBLEM FORMULATION

Without loss of generality, we take two modalities as a showcase to elaborate on the query shift problem. Let $f_{\Theta_s}$ denote the multimodal model pre-trained on the source-domain data $\mathcal{D}_S$, which consists of two modality-specific encoders, *i.e.*, $f_{\Theta_s^Q}$ and $f_{\Theta_s^G}$. For a given query $\mathbf{x}_i^Q$ from the target domain data $\mathcal{D}_T = \left\{ \mathbf{X}^Q = \{\mathbf{x}_i^Q\}_{i=1}^{N^Q}, \mathbf{X}^G = \{\mathbf{x}_j^G\}_{j=1}^{N^G} \right\}$, cross-modal retrieval aims to associate the corresponding sample $\mathbf{x}_j^G$ from the gallery set $\mathbf{X}^G$ by resorting to the common space established by $f_{\Theta_s}$, where $Q$ and $G$ denote as query modality and gallery modality for clarity in the following. In real-world applications, the given queries are usually from the distribution distinct with the source-domain data, *i.e.*, $\mathcal{P}(\mathcal{D}_T) \not\sim \mathcal{P}(\mathcal{D}_S)$, where $\mathcal{P}(\mathcal{D})$ denotes the distribution of the given data $\mathcal{D}$. As

a result, such a query shift problem would lead to the performance degradation of $f_{\Theta_s}$ as verified in the experiments.

To achieve online adaptation for $f_{\Theta_s}$ with query shift, we first propose a simple baseline that endows the unimodal-recognition-oriented TTA approaches with the capacity to handle the cross-modal retrieval task. To be specific, we formulate the retrieval task as a query prediction process in analogy with the recognition task that assigns the given samples to their corresponding categories. Formally, for a given online batch of queries with size $B$, the corresponding query predictions could be defined as

$$\mathbf{p} = \mathrm{Softmax}\left(\mathbf{z}^Q \left(\mathbf{Z}^G\right)^T /\tau\right), \tag{1}$$

where $\tau$ is the temperature that controls the trade-off between the smoothness and sharpness of the predictions, $\mathbf{z}^Q = f_{\Theta_s^Q}(\mathbf{x}^Q) \in \mathbb{R}^{B \times D}$ and $\mathbf{Z}^G = f_{\Theta_s^G}(\mathbf{X}^G) \in \mathbb{R}^{N^G \times D}$ are the $\ell 2$-normalized embeddings with the dimensionality of $D$ for the given query and gallery samples, respectively. Thanks to the above formulation, most existing TTA methods could be adopted to handle the query shift challenge with the following objective,

$$\min_{\tilde{\Theta}} \mathcal{L}_{TTA}\left(\mathbf{p}\right), \tag{2}$$

where $\tilde{\Theta} \subseteq \Theta_s$ is the learnable parameters. However, such a simple formulation of query prediction would result in either model underfitting or overfitting even with carefully tuning of $\tau$ due to the variable gallery sizes, as verified in Table 3 and Fig. 4(a). Furthermore, such a simple baseline overlooks the underlying influences behind the query shift problem and thus fails to achieve promising performance as discussed in Introduction. On the one hand, the existing TTA methods cannot explicitly manipulate the intra-modality uniformity and inter-modality gap. On the other hand, these methods struggle to account for the heavy noise from the query prediction.

As a remedy, we propose Test-time adaptation for Cross-modal Retrieval (TCR). As shown in Fig. 2, for the given online batch of queries, TCR first employs the novel query prediction refinement module to obtain the retrieval-favorable predictions $\hat{\mathbf{p}}$. After that, the following joint objective consisting of three novel loss functions is adopted to achieve robust cross-modal retrieval against the query shift, *i.e.*,

$$\min_{\tilde{\Theta}} \mathcal{L}\left(\hat{\mathbf{p}}\right), \tag{3}$$

where $\mathcal{L} = \mathcal{L}_{MU} + \mathcal{L}_{MG} + \mathcal{L}_{NA}$, with $\mathcal{L}_{MU}, \mathcal{L}_{MG}$, and $\mathcal{L}_{NA}$ denote the intra-modality uniformity learning loss, inter-modality gap learning loss, and the noise-robust adaptation loss, respectively. In the following, we will elaborate on each loss individually.

## 3.2 QUERY PREDICTION REFINEMENT

In this section, we introduce the query prediction refinement module which involves refining the prediction for the given queries and estimating the constraints to support the optimization of Eq. 3.

### 3.2.1 CANDIDATE SELECTION

To break the dilemma of vanilla query prediction formulation in Eq. 1, we propose selecting a subset of candidates from the gallery and then establishing new query predictions for the given queries. Mathematically, for each given query $\mathbf{x}_i^Q$, the corresponding candidate in the gallery is obtain via

$$\mathbf{x}_i^{G'} = \mathcal{N}(\mathbf{x}_i^Q), \tag{4}$$

where $\mathcal{N}(\cdot)$ denotes the selection manner, and we adopt the nearest neighborhood selection in the common space for simplicity. In other words, we retrieve the most similar sample from the gallery set for each query and thus obtain query-candidate pairs $(\mathbf{z}^Q, \mathbf{z}^{G'})$. Consequently, the refined query predictions for the online-batched queries $\mathbf{x}^Q$ could be formulated as follows,

$$\hat{\mathbf{p}} = \mathrm{Softmax}\left(\mathbf{z}^Q \left(\mathbf{Z}^{G'}\right)^T /\tau\right), \tag{5}$$

where $\mathbf{Z}^{G'}$ are the embeddings of the selected candidates for $\mathbf{x}^Q$. The query prediction refinement manner embraces the following two merits. On the one hand, the query prediction refinement manner could exclude some irrelevant samples in the gallery, thus preventing the model from overfitting

to some extent. On the other hand, the excluded irrelevant samples would avoid looking for a needle in a bottle of hay for queries, thus alleviating the model underfitting issue.

### 3.2.2 CONSTRAINT ESTIMATION

It is widely acknowledged that source data could effectively regulate the domain adaptation process (Long et al., 2017; Kang et al., 2019), thus circumventing the catastrophic forgetting issue. Due to the unavailability of the source data in the test-time adaptation, we propose further choosing some source-domain-like data from the selected query-candidate pairs to estimate desirable constraints that support the optimization of Eq. 3. To this end, we first design a criterion for choosing the query-candidate pair $(\mathbf{z}_i^Q, \mathbf{z}_i^{G'})$, i.e.,

$$\mathrm{SI} = 2\left(\|\mathbf{z}_i^Q - \mathbf{z}_i^{G'}\|\right) - \left(\|\mathbf{z}_i^Q - \overline{\mathbf{z}}^Q\| + \|\mathbf{z}_i^{G'} - \overline{\mathbf{z}}^{G'}\|\right),\tag{6}$$

where $\overline{\mathbf{z}}^Q = \frac{1}{B}\sum_i^B \mathbf{z}_i^Q$ and $\overline{\mathbf{z}}^{G'} = \frac{1}{B}\sum_i^B \mathbf{z}_i^{G'}$ are the centers of the given queries and the selected candidates, respectively. In the implementation, we choose $30\%$ query-candidate pairs with the smallest SI value, namely, $(\mathbf{z}^{Q_m}, \mathbf{z}^{G'_m})$ of size $M$, as the source-domain-like data. Clearly, a low value of the criterion has the incentive to select the query-candidate pairs with the small modality gap and high intra-modality uniformity, which have higher probability to be source-domain-like, as verified in Fig. 3.

Based on the selected source-domain-like data, we propose estimating the modality gap of the source model as follows,

$$\Delta_S = \left\|\frac{1}{M}\sum_i^M \mathbf{z}_i^{Q_m} - \frac{1}{M}\sum_j^M \mathbf{z}_j^{G'_m}\right\|,\tag{7}$$

where $\mathbf{z}_i^{Q_m}$ and $\mathbf{z}_j^{G'_m}$ denote the $i$-th query sample and the $j$-th gallery sample in the query-candidate pairs, respectively. As the another by-product, a desirable threshold that could filter the noise in the query predictions $\hat{\mathbf{p}}$ could be adaptively determined as follows,

$$E_m = \max_{i=1,\ldots,M} E\left(\mathbf{x}_i^{Q_m}\right),\tag{8}$$

where $E\left(\cdot\right)$ indicates the entropy based on the refined query predictions.

### 3.3 INTRA-MODALITY UNIFORMITY LEARNING

As discussed in Introduction, query shift would diminish the uniformity of the query modality, resulting in confused queries with lower discrimination in the common space. To address the problem, we propose to perform the contrast between queries and their respective centers, thus explicitly enlarging the intra-modality uniformity. Mathematically, the loss function of intra-modality uniformity learning is defined as follows,

$$\mathcal{L}_{MU} = \frac{1}{B}\sum_i^B exp\left(-\|\mathbf{z}_i^Q - \overline{\mathbf{z}}^Q\|/t\right),\tag{9}$$

where $t$ is the trade-off parameter to control the uniformity that is fixed as $10$ in the experiments.

### 3.4 INTER-MODALITY GAP LEANING

As discussed in Introduction, query shift would amplify the modality gap between query and gallery modalities, disrupting the cross-modal alignment established by the source model. To remedy this, we propose to rectify the difference between the query and gallery modalities to the estimated modality gap of the source model in Eq. 7. Formally, the inter-modality gap learning loss is defined as follows,

$$\mathcal{L}_{MG} = \left(\Delta_T - \Delta_S\right)^2,\tag{10}$$

where $\Delta_T = \left\|\overline{\mathbf{z}}^Q - \overline{\mathbf{z}}^{G'}\right\|$ denotes the modality gap of the target domain. The key idea behind $\mathcal{L}_{MG}$ is that the modality gap rectification would take advantage of well-aligned multimodal common

space from the source model, thus boosting retrieval performance. Notably, as observed in Liang et al. (2022), over-eliminating the modality gap would not improve or even degrade the performance of the multimodal model. Therefore, we believe the proposed loss that rectifies the modality gap of the target model to a plausible constraint in a non-monotonic manner is reasonable. The experimental results in Fig. 3 could support the claims.

## 3.5 NOISE-ROBUST ADAPTATION

To achieve robustness against the heavy noise on the query predictions, we propose the following noise-robust adaptation loss,

$$
\mathcal{L}_{NA} = \frac{1}{\sum_i \mathbb{I}_{\{S(\mathbf{x}_i^Q) \neq 0\}}} \sum_{i=1}^{N^Q} S(\mathbf{x}_i^Q) E(\mathbf{x}_i^Q), \text{ where } S(\mathbf{x}_i^Q) = \max\left(1 - \frac{E(\mathbf{x}_i^Q)}{E_m}, 0\right), \quad (11)
$$

where $E_m$ is the self-adaptive threshold estimated in Eq. 8, and $\mathbb{I}_{\{\cdot\}}$ is an indicator function evaluating to 1 *i.f.f.* the condition is satisfied. Such behavior could achieve robustness against noise by excluding high-entropy query predictions from adaptation and assigning higher weights to query predictions with lower uncertainties.

It is worth noting that existing TTA methods like EATA (Niu et al., 2022) and SAR (Niu et al., 2023) are highly sensitive to the manually determined thresholds, resulting in either none or all query predictions being treated as noise. In contrast, the proposed $\mathcal{L}_{NA}$ employs a self-adaptive threshold to filter out noise, thus achieving better performance.

## 4 EXPERIMENTS

In this section, we verify the effectiveness of TCR in handling the query shift problem for the image-text retrieval task. This section is organized as follows. In Section 4.1, we present the implementation details and experiment settings of TCR. In Section 4.2, we conduct extensive comparison experiments to verify the performance superiority of TCR. In Section 4.3, we perform a series of analytic studies, ablation studies, and visualization analyses, to provide a comprehensive understanding of TCR.

## 4.1 IMPLEMENTATION DETAILS AND EXPERIMENT SETTINGS

TCR is a general TTA framework that could endow most existing pre-trained models with robustness against the query shift. Therefore, we select CLIP (Radford et al., 2021) and BLIP (Li et al., 2022) as the source models since they are the widely adopted vision-language models for image-text retrieval task. Following Lee et al. (2018), we adopt two testing protocols, namely, image-to-text retrieval (a.k.a. TR) and text-to-image retrieval (a.k.a. IR). During the adaptation process, TCR performs the objective function for each coming mini-batch of queries, and the batch size is set as 64. Following Niu et al. (2023); Wang et al. (2021), TCR updates the parameters within the normalization layers in the query-specific encoder $f_{\Theta_s^Q}$ using the AdamW optimizer. To be more specific, the learnable parameters in $\tilde{\Theta}$ (Eq. 3) correspond to the Layer Normalization (LN) layers in our implementation. Besides, the temperature hyper-parameter $\tau$ in Eq. 1 and uniformity learning hyper-parameter $t$ in Eq. 9 are fixed as 0.02 and 10 for all experiments, respectively.

To investigate the influence of cross-modal retrieval with query shift, we employ the following two settings for extensive evaluations (see more details in Appendix B.1).

- **Query Shift** (QS): In this setting, only the queries come from different distributions with the source-domain data. Following Qiu et al. (2023), we introduce 16 types of corruptions to the image modality and 15 types to the text modality across widely-used image-text retrieval datasets, COCO (Lin et al., 2014) and Flickr (Plummer et al., 2015). As a result, the COCO-C and Flickr-C benchmarks are constructed, which would result in distribution shifts on either the image or text modalities. To guarantee the controlled study on QS, we first fine-tune the pre-trained model on either the COCO (Lin et al., 2014) or Flickr (Plummer et al., 2015) dataset, namely, treating them as the source domains. After that, evaluations are conducted on the COCO-C or Flickr-C benchmarks, namely, treating them as the target domain.

- **Query-Gallery Shift** (QGS): In this setting, both the query and gallery samples are drawn from distributions different from the source-domain data. To this end, evaluations are directly conducted on the pre-trained model upon several widely-used image-text retrieval datasets from various domains, including Fashion-Gen (Rostamzadeh et al., 2018) from the e-commerce domain, CUHK-PEDES (Li et al., 2017) and ICFG-PEDES (Ding et al., 2021) from the person re-identification (ReID) domain, and COCO, Flickr, and Nocaps (Agrawal et al., 2019) from the natural image domain. In other words, the source model would encounter distribution shifts on both image and text modalities during adaptation.

Table 1: Comparisons with state-of-the-art methods on COCO-C benchmark under **QUERY SHIFT ON THE IMAGE MODALITY** with maximum severity level regarding the Recall@1 metric. The best results are marked in **bold**.

| Query Shift | Noise | | | | Blur | | | | Weather | | | | Digital | | | | Avg. |
|---|---|---|---|---|---|---|---|---|---|---|---|---|---|---|---|---|---|
| | Gauss. | Shot | Impul. | Speckle | Defoc. | Glass | Motion | Zoom | Snow | Frost | Fog | Brit. | Contr. | Elastic | Pixel | JPEG | |
| BLIP ViT-B/16 | 43.4 | 46.3 | 43.2 | 57.3 | 43.3 | 68.0 | 39.7 | 8.4 | 32.3 | 52.2 | 57.0 | 66.8 | 36.0 | 41.3 | 20.6 | 63.7 | 45.0 |
| • Tent | 41.6 | 40.5 | 37.9 | 54.0 | 44.7 | 65.1 | 39.6 | 8.3 | 31.9 | 48.7 | 56.3 | 66.5 | 31.8 | 40.3 | 19.2 | 62.3 | 43.0 |
| • EATA | 41.4 | 50.3 | 35.7 | 63.1 | 49.8 | 72.2 | 46.2 | 6.9 | 45.6 | 56.7 | 62.5 | 71.4 | 43.6 | 51.3 | 25.6 | 67.0 | 49.3 |
| • SAR | 42.3 | 51.5 | 37.5 | 61.8 | 40.3 | 71.5 | 32.8 | 6.2 | 38.0 | 56.2 | 59.1 | 70.6 | 31.1 | 53.5 | 17.5 | 66.4 | 46.0 |
| • READ | 45.8 | 48.4 | 37.2 | 59.9 | 44.5 | 71.8 | 46.6 | 11.5 | 39.9 | 49.9 | 58.4 | 70.3 | 35.8 | 45.0 | 18.8 | 66.2 | 46.9 |
| • DeYO | 47.9 | 53.5 | 46.8 | 63.4 | 42.9 | 72.1 | 36.7 | 3.2 | 37.5 | 59.7 | 66.4 | 71.2 | 40.3 | 49.0 | 13.1 | 67.6 | 48.2 |
| • Ours | **53.2** | **56.2** | **54.8** | **64.6** | **58.0** | **73.7** | **56.4** | **32.2** | **56.5** | **64.1** | **71.0** | **73.4** | **57.9** | **63.7** | **41.8** | **68.4** | **59.1** |
| BLIP ViT-L/16 | 50.3 | 51.8 | 51.1 | 61.6 | 53.7 | 72.1 | 49.4 | 14.5 | 44.0 | 57.5 | 61.8 | 70.5 | 37.3 | 50.6 | 32.0 | 70.5 | 51.8 |
| • Tent | 46.3 | 49.3 | 46.7 | 58.4 | 52.2 | 71.8 | 47.5 | 12.3 | 41.9 | 56.2 | 60.9 | 69.7 | 35.7 | 48.3 | 29.4 | 69.6 | 49.8 |
| • EATA | 46.2 | 53.5 | 49.5 | 63.8 | 56.5 | 73.8 | 47.2 | 18.4 | 50.6 | 59.1 | 64.5 | 72.1 | 40.7 | 55.4 | 43.5 | 70.7 | 54.4 |
| • SAR | 45.9 | 50.2 | 47.3 | 63.1 | 51.1 | 73.8 | 47.2 | 11.6 | 40.8 | 58.9 | 60.7 | 71.6 | 33.6 | 54.0 | 34.4 | 70.5 | 50.9 |
| • READ | 38.1 | 48.0 | 43.3 | 63.5 | 43.6 | 73.4 | 43.6 | 22.0 | 44.5 | 56.5 | 62.2 | 71.9 | 32.9 | 49.6 | 27.5 | 70.6 | 49.5 |
| • DeYO | 39.9 | 50.2 | 43.5 | 63.8 | 50.4 | 74.0 | 52.4 | 5.4 | 49.5 | 59.3 | 62.8 | 71.8 | 34.0 | 54.7 | 34.4 | 69.7 | 51.0 |
| • Ours | **58.2** | **60.7** | **59.8** | **66.6** | **61.5** | **74.9** | **60.3** | **36.8** | **59.0** | **65.2** | **72.1** | **73.5** | **56.3** | **65.7** | **50.2** | **71.6** | **62.0** |

Table 2: Comparisons with state-of-the-art methods on COCO-C benchmark under **QUERY SHIFT ON THE TEXT MODALITY** with maximum severity level regarding the Recall@1 metric.

| Query Shift | Character-level | | | | | Word-level | | | | | Sentence-level | | | | | Avg. |
|---|---|---|---|---|---|---|---|---|---|---|---|---|---|---|---|---|
| | OCR | CI | CR | CS | CD | SR | RI | RS | RD | IP | Formal | Casual | Passive | Active | Backtrans | |
| BLIP ViT-B/16 | 31.4 | 11.3 | 9.4 | 18.9 | 11.4 | 43.6 | 51.5 | 50.3 | 50.6 | 56.8 | 56.6 | 56.2 | 54.9 | 56.8 | 54.2 | 40.9 |
| • Tent | 31.4 | 11.0 | 9.5 | 17.7 | 11.3 | 43.2 | 51.3 | 50.3 | 50.6 | 56.6 | 56.2 | 56.0 | 54.9 | 56.9 | 53.9 | 40.7 |
| • EATA | 33.1 | 11.9 | 10.5 | 18.4 | 12.0 | 44.9 | 53.0 | 51.6 | 50.3 | 56.2 | 56.8 | **56.8** | **56.0** | 56.8 | 54.3 | 41.5 |
| • SAR | 31.8 | 11.6 | 9.9 | 18.5 | 11.7 | 43.6 | 51.5 | 50.3 | 50.6 | 56.8 | 56.5 | 56.2 | 54.9 | 56.8 | 54.2 | 41.0 |
| • READ | 32.3 | 11.4 | 9.6 | 18.2 | 11.2 | 44.3 | 52.9 | 51.7 | 51.1 | 57.6 | 57.1 | 56.7 | 55.9 | 57.1 | **54.7** | 41.4 |
| • DeYO | 31.4 | 11.3 | 9.4 | 17.9 | 11.4 | 43.6 | 51.5 | 50.3 | 50.6 | 56.8 | 56.5 | 56.2 | 54.9 | 56.7 | 54.2 | 40.9 |
| • Ours | **34.1** | **13.7** | **11.8** | **19.5** | **13.2** | **45.3** | **53.8** | **51.8** | **51.5** | **57.3** | **57.1** | **56.8** | **56.0** | **57.3** | **54.7** | **42.3** |
| BLIP ViT-L/16 | 34.5 | 12.3 | 11.1 | 19.7 | 12.9 | 46.0 | 54.4 | 54.0 | 53.5 | 59.4 | 59.1 | 58.8 | 57.8 | 59.4 | 56.7 | 43.3 |
| • Tent | 34.0 | 12.3 | 11.0 | 19.6 | 12.9 | 46.5 | 54.2 | 53.8 | 53.4 | 59.4 | 59.1 | 58.8 | 57.6 | 58.9 | 56.5 | 43.2 |
| • EATA | 35.6 | 13.3 | 11.3 | 20.3 | 13.2 | 47.2 | 55.4 | 54.2 | 53.8 | 59.2 | 59.1 | 59.4 | 57.9 | 59.4 | 56.7 | 43.7 |
| • SAR | 34.5 | 13.1 | 11.2 | 20.3 | 13.1 | 46.7 | 54.4 | 54.0 | 53.5 | 59.5 | 59.1 | 58.8 | 57.8 | 59.4 | 56.7 | 43.5 |
| • READ | 35.3 | 12.2 | 10.9 | 19.1 | 12.7 | 47.3 | 55.1 | 55.0 | 53.3 | 59.7 | 59.3 | **59.1** | 58.1 | **59.6** | 56.7 | 43.6 |
| • DeYO | 34.5 | 12.3 | 11.1 | 19.7 | 12.9 | 46.7 | 54.4 | 54.0 | 53.5 | 59.5 | 59.1 | 58.8 | 57.8 | 59.4 | 56.7 | 43.4 |
| • Ours | **36.8** | **14.7** | **13.4** | **21.3** | **14.3** | **47.9** | **56.3** | **54.8** | **53.9** | **59.5** | **59.4** | 59.0 | **58.2** | **59.6** | **56.9** | **44.4** |

## 4.2 COMPARISONS WITH STATE-OF-THE-ARTS

In this section, We compare TCR with five SOTA TTA methods (Tent (Wang et al., 2021), EATA (Niu et al., 2022), SAR (Niu et al., 2023), READ (Yang et al., 2024), and DeYO (Lee et al., 2024)) under both the QS and QGS settings. Among the baseline methods, Tent is the vanilla TTA approach with an entropy-based objective, while the others enhance Tent by incorporating specially designed noise-robust loss functions. For a fair comparison, we select the optimal temperature (Eq.1) for the TTA baselines upon each dataset according to Fig. 4(a). The results on the QS setting and QGS setting are summarized in Tables 1-2, 7-8 and Tables 3-4, respectively. From the results, one could have the following observations and conclusions.

- Existing TTA methods only achieve marginal performance improvements over the base model, which could be attributed to the inability on manipulating both modality uniformity and the modality gap. In contrast, TCR could rectify the modality gap and enlarge the modality uniformity, thus significantly outperforming all the baselines across various pre-trained model types and sizes.
- TCR demonstrates greater robustness against more severe shift types like "Zoom" and "Pixel" (see Table 1 and Table 7), whereas most baseline methods experience significant performance degradation under these challenging distribution shifts. Moreover, in Table 3,

the more significant performance improvements on "Base2Nocaps" with "ND" and "OD" compared to that with "ID" also verify the conclusion.

- As the size of the gallery set increases in Table 3 (from "Base2Flickr" to "Base2COCO" to "Base2Fashion"), existing TTA methods suffer from increasing performance degradation, eventually performing even worse than the base model. This supports our claim in Section 3.1 that TTA methods struggle to accommodate well for cross-modal retrieval tasks due to the excessively large gallery set. In contrast, our method consistently achieves performance improvements across various gallery sizes.

Table 3: Comparisons with state-of-the-art methods on benchmarks under **QUERY-GALLERY SHIFT** regarding the Recall@1 metric. In the table, "ID", "ND" and "OD" refer to "In-Domain", "Near-Domain" and "Out-Domain", respectively. Besides, "TR@1" / "IR@1" represent Recall@1 for image-to-text retrieval / text-to-image retrieval.

| | Base2Flickr | | Base2COCO | | Base2Fashion | | Base2Nocaps(ID) | | Base2Nocaps(ND) | | Base2Nocaps(OD) | | |
| Query Shift | TR@1 | IR@1 | TR@1 | IR@1 | TR@1 | IR@1 | TR@1 | IR@1 | TR@1 | IR@1 | TR@1 | IR@1 | Avg. |
|---|---|---|---|---|---|---|---|---|---|---|---|---|---|
| CLIP ViT-B/16 | 80.2 | 61.5 | 52.5 | 33.0 | 8.5 | 13.2 | 84.9 | 61.4 | 75.4 | 49.2 | 73.8 | 55.8 | 54.1 |
| • Tent | 81.4 | 64.0 | 48.8 | 27.6 | 5.6 | 10.7 | 85.1 | 61.7 | 74.6 | 48.6 | 71.8 | 56.1 | 53.0 |
| • EATA | 80.4 | 63.4 | 52.1 | 34.8 | 8.1 | 12.0 | 84.7 | 62.0 | 75.1 | 52.3 | 74.1 | 56.9 | 54.7 |
| • SAR | 80.3 | 62.2 | 51.8 | 33.9 | 8.0 | 13.3 | 84.7 | 61.3 | 75.4 | 51.3 | 73.7 | 56.1 | 54.3 |
| • READ | 80.6 | 64.4 | 46.0 | 35.7 | 5.8 | 11.2 | 85.1 | 63.0 | 75.0 | 52.1 | 73.5 | 57.0 | 54.1 |
| • DeYO | 80.1 | 64.0 | 51.5 | 33.4 | 6.9 | 10.9 | 84.4 | 62.2 | 75.1 | 52.0 | 73.2 | 57.3 | 54.3 |
| • Ours | **82.4** | **64.8** | **52.9** | **36.5** | **8.9** | **14.0** | **85.1** | **63.5** | **75.7** | **54.0** | **74.4** | **58.0** | **55.9** |
| BLIP ViT-B/16 | 70.0 | 68.3 | 59.3 | 45.4 | 19.9 | 26.1 | 88.2 | 74.9 | 79.3 | 63.6 | 81.9 | 67.8 | 62.1 |
| • Tent | 81.9 | 68.5 | 61.7 | 41.7 | 14.1 | 26.1 | 88.5 | 75.4 | 82.6 | 64.1 | 82.7 | 68.9 | 63.0 |
| • EATA | 82.3 | 69.4 | 64.2 | 47.9 | 12.8 | 25.2 | 87.8 | 75.1 | 82.8 | 63.9 | 81.5 | 67.9 | 63.4 |
| • SAR | 81.7 | 68.3 | 63.5 | 46.6 | 17.9 | 26.1 | 88.2 | 75.6 | 81.0 | 65.4 | 81.2 | 69.3 | 63.7 |
| • READ | 80.0 | 69.9 | 62.1 | 46.4 | 5.6 | 24.1 | 87.3 | 75.1 | 80.6 | 63.9 | 80.7 | 67.9 | 62.0 |
| • DeYO | 83.5 | 69.9 | 65.0 | 47.3 | 12.2 | 24.1 | 89.2 | 75.6 | 83.7 | 65.7 | 84.3 | 69.4 | 64.2 |
| • Ours | **86.8** | **70.3** | **68.9** | **48.9** | **23.6** | **30.3** | **89.7** | **76.0** | **86.3** | **66.1** | **87.2** | **69.5** | **67.0** |

### 4.3 ABLATION AND ANALYTIC STUDY

In this section, all the experiments are conducted under the "Base2COCO" setting using the BLIP ViT-B/16 model unless otherwise stated.

**Analytic Studies on Intra-modality Uniformity and Inter-modality Gap.** As pointed out in Introduction, the query shift would diminish the intra-modality uniformity and amplify the inter-modality gap. For an in-depth understanding, we conduct analytic experiments to investigate how the two characteristics affect the retrieval performance. To examine the influence of the intra-modality uniformity, we manually scale the latent distance between different queries. Mathematically, the scaling operation is defined as follows.

$$(\mathbf{z}_i^Q)^{\text{scale}} = \overline{\mathbf{Z}}^Q + \lambda^{\text{scale}} \left( \mathbf{z}_i^Q - \overline{\mathbf{Z}}^Q \right), \tag{12}$$

where $\overline{\mathbf{Z}}^Q = \frac{1}{N^Q} \sum_{i=1}^{N^Q} \mathbf{z}_i^Q$ is the center of queries, $\lambda^{\text{scale}}$ is the scaling factor. As illustrated in Fig. 3(a), increasing the intra-modality uniformity in the query modality would improve the performance, but not vice versa. Such a phenomenon indicates that higher uniformity would guarantee the discrimination between queries and thus boost performance. To examine the influence of the inter-modality gap, following Liang et al. (2022), we manually move every query embedding towards closing the modality gap. Formally,

$$(\mathbf{z}_i^Q)^{\text{offset}} = \mathbf{z}_i^Q - \lambda^{\text{offset}} \left( \overline{\mathbf{Z}}^Q - \overline{\mathbf{Z}}^G \right), \tag{13}$$

Table 4: Comparisons with state-of-the-art methods on ReID benchmarks under **QUERY-GALLERY SHIFT** regarding the Recall@1 metric.

| | CUHK2ICFG | ICFG2CUHK | |
| Query Shift | IR@1 | IR@1 | Avg. |
|---|---|---|---|
| CLIP ViT-B/16 | 33.3 | 41.0 | 37.2 |
| • Tent | 33.5 | 41.9 | 37.7 |
| • EATA | 33.3 | 42.2 | 37.8 |
| • SAR | 33.3 | 42.2 | 37.8 |
| • READ | 33.0 | 42.3 | 37.7 |
| • DeYO | 33.3 | 42.2 | 37.8 |
| • Ours | **37.3** | **42.4** | **39.9** |

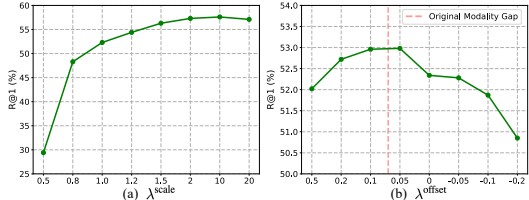

Figure 3: Observation of the intra-modality uniformity and inter-modality gap. The increasing $\lambda^{\text{scale}}$ indicates the growing intra-modality uniformity while the decreasing $\lambda^{\text{offset}}$ indicates the narrowing inter-modality gap. Notably, $\lambda^{\text{scale}} = 1.0$ and $\lambda^{\text{offset}} = 0$ represent no scaling and no offset, respectively.

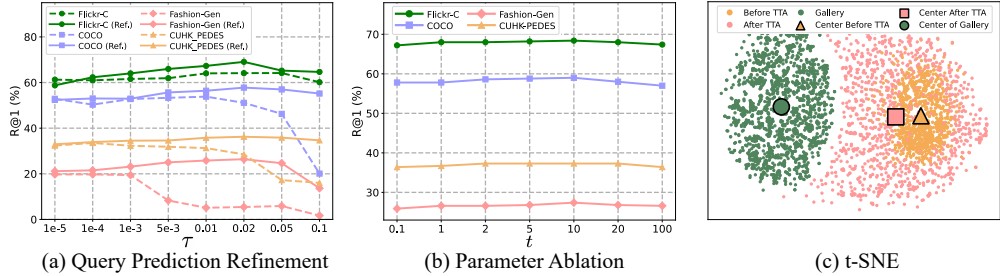

(a) Query Prediction Refinement  (b) Parameter Ablation  (c) t-SNE

Figure 4: Finer-grained Ablation studies. (a) The parameter analysis of $\tau$ (Eq. 1 and Eq. 5) on the vanilla TTA method Tent w/ (solid line) and w/o (dotted line) the query prediction refinement module. (b) The parameter analysis of $t$ in Eq. 9. (c) The t-SNE visualization of TR on the query and gallery embeddings after employing the proposed TCR.

where $\overline{\mathbf{Z}}^G = \frac{1}{N^G} \sum_{i=1}^{N^G} \mathbf{z}_i^G$ is the center of the gallery samples, $\lambda^{\text{offset}}$ controls the offset of the embeddings. From the results in Fig. 3(b), one could observe that monotonously eliminating the modality gap would not always improve the performance. In contrast, the estimated modality gap from the pre-trained model is a plausible criterion for modality gap rectification. Note that the embeddings are all $\ell 2$-normalized after scaling or shifting.

**Ablation studies.** To verify the effectiveness of each design, we investigate the loss terms of TCR in Table 5, resulting in the following conclusions. First, $\mathcal{L}_{NA}$ boosts performance through the query prediction refinement module and self-adaptive loss, *e.g.*, TCR improves R@1 by 9.2% and 14.6% on text and image retrieval, compared to Tent in Table 3. Second, both the designed intra-modality uniformity learning module and inter-modality gap learning module would enhance robustness against query shift. Third, TCR achieves optimal performance when all the loss terms are employed. Moreover, we carry out experiments to

Table 5: Ablation study of the loss functions, where " $\checkmark$ " denotes the loss is adopted.

| $\mathcal{L}_{NA}$ | $\mathcal{L}_{MU}$ | $\mathcal{L}_{EMG}$ | TR@1 | IR@1 | Avg. |
|---|---|---|---|---|---|
| | | | 59.3 | 45.4 | 52.4 |
| $\checkmark$ | | | 67.4 | 47.8 | 57.9 |
| | $\checkmark$ | | 64.9 | 46.7 | 55.8 |
| | | $\checkmark$ | 64.3 | 46.3 | 55.3 |
| $\checkmark$ | $\checkmark$ | | 67.8 | 48.3 | 58.2 |
| $\checkmark$ | | $\checkmark$ | 68.1 | 48.4 | 58.4 |
| | $\checkmark$ | $\checkmark$ | 66.3 | 47.8 | 57.1 |
| $\checkmark$ | $\checkmark$ | $\checkmark$ | 68.9 | 48.9 | 58.9 |

verify the effectiveness of the proposed query prediction refinement module in Section 3.5. As shown in Fig. 4(a), we observe that: i) selecting an appropriate temperature for the existing TTA approach across various datasets is challenging; ii) even a low temperature (*e.g.*, $1e-4$) is a better setting across all datasets, the performance degrades as a low temperature tends to make model overfitting on noisy query prediction. In contrast, the query prediction refinement module not only stabilizes the temperature setting for all the datasets but also prevents the model from either underfitting or overfitting by excluding some irrelevant samples in the gallery. Besides, TCR demonstrates stable performance within the range of [0.001,0.05] and achieves the best performance when $\tau = 0.02$. As depicted in Fig. 4(b), one could observe that TCR is not sensitive to the choice of $t$.

**Visualization Result.** To qualitatively study the effectiveness of TCR, we conduct the t-SNE visualization on both the query and gallery embeddings before and after the TTA process. From the results in Fig. 4(c), one could observe that the samples in the query modality enjoy more scatter and the difference between the query and gallery modalities narrows after the TTA process. In other words, TCR achieves better robustness against query shift by rectifying the intra-modality uniformity and the inter-modality gap.

## 5 CONCLUSION

In this paper, we develop a new test-time adaptation method, dubbed TCR, to achieve robust cross-modal retrieval against the query shift problem. In brief, TCR first employs a novel module to refine the query predictions. After that, TCR adopts a novel joint objective to prevent the model adaptation from overfitting noise while simultaneously manipulating the intra-modality uniformity and inter-modality gap to preserve the well-established common space from the source model. Extensive experiments not only verify the effectiveness of TCR but also reveal the importance of each designs. In the future, we plan to explore more potential scenarios contaminated with query shift and extend TCR to address the corresponding issues.

## ACKNOWLEDGMENTS

This work was supported in part by NSFC under Grant 62176171, U21B2040, 624B2099, 623B2075, 62472295; in part by the Fundamental Research Funds for the Central Universities under Grant CJ202303; and in part by Sichuan Science and Technology Planning Project under Grant 24NSFTD0130.

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

# APPENDIX

## CONTENTS

## A  DEFINATION OF INTRA-MODALITY UNIFORMITY AND INTER-MODALITY GAP

Here, we provide a mathematical definition of intra-modality uniformity and inter-modality gap, two essential characteristics in multimodal learning (Li et al., 2023a; Lu et al., 2024). Specifically, the modality uniformity of query modality is defined as

$$\text{Uniformity} = \frac{1}{N^Q} \sum_{i=1}^{N^Q} \|\mathbf{z}_i^Q - \overline{\mathbf{Z}}^Q\|. \tag{14}$$

A low intra-modality uniformity illustrates that the samples in the query modality are compact, which degrades the retrieval performance as the model struggles to distinguish diverse queries.

The modality gap between query and gallery modalities is defined as

$$\text{Modality Gap} = \|\overline{\mathbf{Z}}^Q - \overline{\mathbf{Z}}^G\|. \tag{15}$$

Modality gap has been proved to be an inherent characteristic of multimodal pre-trained models in Liang et al. (2022), which might represent the well-aligned common space to some extent. As shown in Fig. 4(a), either an over-low or over-high modality gap would destroy the well-constructed common space, harming the retrieval performance.

## B  MORE IMPLEMENTATION DETAILS

### B.1  MORE DETAILS ABOUT THE BENCHMARKS

In the manuscript, we employ the QS setting and the QGS setting for evaluation. Here, we provide more detail about the benchmarks employed in the two settings.

**Query Shift.** We construct two benchmarks with only query modality distribution shifts based on the widely-used COCO and Flickr datasets. Specifically,

- COCO is a large-scale dataset for cross-modal retrieval and image captioning tasks. For evaluation, we conduct experiments on the COCO 2014 testing set following Li et al. (2022), which contains 5,000 images and 25,000 annotations, with each image associated with five corresponding text descriptions.
- Flickr is a cross-modal retrieval dataset collected from natural scenarios. Following Radford et al. (2021), we employ the test set comprising 1,000 images and 5,000 annotations, where each image is paired with five corresponding sentences.

Following Qiu et al. (2023), we introduce 16 and 15 types of corruption to the image and text modality, respectively. Specifically, the corruptions in image modality consist of: (1) Noise: Gaussian noise, Shot noise, Impulse noise, Speckle noise; (2) Blur: Defocus blur, Glass blur, Motion blur, Zoom blur ; (3) Weather: Snow, Frost, Fog, Brightness; (4) Digital: Contrast, Elastic, Pixelate, JPEG compression. To simulate real-world corruptions, each image modality corruption is applied at five different severity levels, resulting in a total of 80 perturbations. As for the text modality, the employed corruptions could be categorized into three levels: character-level, word-level, and sentence-level. Specifically, the character-level corruptions consist of OCR, Character Insert (CI), Character Replace (CR), Character Swap (CS), and Character Delete (CD), which simulate real-world typos or mistakes during typing. The word-level corruptions involve Synonym Replacement (SR), Word Insertion (WR), Word Swap (WS), Word Deletion (WD), and Insert Punctuation (IP), which simulate different writing habits that people may replace, delete, or add words to express the same meaning. For sentence-level corruptions, we convert the annotation styles into Formal, Casual, Passive, Active, and Back-translation, which simulate various speaking, writing styles or translation errors. Similar to the image corruptions, we introduce 7/2/1 severity levels for character-level/word-level/sentence-level corruptions.

As a result, we construct the two benchmarks named COCO-C and Flickr-C. Notably, we only introduce the corruptions to the query modality in the QS setting, *e.g.*, for image-to-text retrieval, the distribution shifts occur on the image modality. The cases of the 16 image corruptions and 15 text corruptions are visualized in Fig. 5 and Table 6.

**Query-Gallery Shift.** We establish the following benchmarks with distribution shifts across both query and gallery modalities, including Fashion-Gen from the E-commerce domain, CUHK-PEDES and ICFG-PEDES from the ReID domain, as well as COCO, Flickr, and Nocaps from the natural image domain. Specifically,

- Fashion-Gen is a cross-modal retrieval dataset source from the E-commerce domain, comprising fashion images paired with item descriptions provided by professional stylists. In the experiment, we employ the testing set containing 32,528 image-text pairs.
- CUHK-PEDES is a text-to-image person re-identification dataset derived from short-duration surveillance videos. Following Jiang & Ye (2023), we utilize the testing set which contains 3,074 images and 6,156 textual descriptions of 1,000 identities. The cases of the CUHK-PEDES are visualized in Fig. 6 (a).
- ICFG-PEDES is a large-scale text-to-image person re-identification dataset gathered at different times of the day (*i.e.*, morning, noon, afternoon). Following Jiang & Ye (2023), we adopt the testing set consists of 19,848 image-text pairs, corresponding to 1,000 identities. The examples of the ICFG-PEDES are visualized in Fig. 6 (b).
- Nocaps is a cross-modal retrieval dataset derived from the OpenImages dataset. For evaluation, we perform experiments on the test set, which consists of 648 in-domain images, 2,938 near-domain images, and 914 out-domain images. Each image is paired with 10 captions.

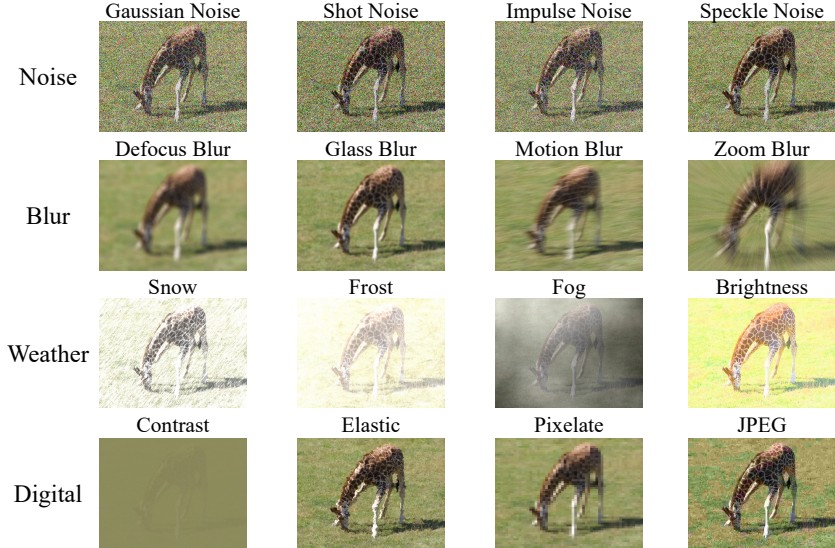

Figure 5: Examples of 16 types of image corruption. The original image is from the COCO dataset.

Table 6: Examples of 15 types of text corruption. The original text is from the COCO dataset.

| Category | Perturbation | Example |
|---|---|---|
| Original | Clean | A train traveling down tracks next to a brick building. |
| Character | OCR | A train travelin9 down track8 next to a brick building. |
| | CI | A train traveling down traGcks next to a brick bui1lding. |
| | CR | A train traveling doPn tracks next to a brick buildirg. |
| | CS | A train rtaveilng down tracks next to a brick building. |
| | CD | A train tr[X]veling down tr[X]cks next to a brick building. |
| Word | SR | A train jaunt down running adjacent to a brick building. |
| | RI | A train pass traveling down tracks next to go a brick building |
| | RS | A building traveling down tracks next to a brick train. |
| | RD | A train [X] down tracks [X] to a brick building. |
| | IP | A : train traveling down tracks next to , a brick building. |
| Sentence | Formal | A train moving down tracks next to a brick building. |
| | Casual | A train that goes down tracks next to a brick building. |
| | Passive | Tracks next to a brick building are being traveled down by a train. |
| | Active | There is a train traveling down tracks next to a brick building. |
| | Backtrans | A train runs down the tracks next to a brick building. |

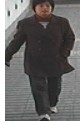 A pale oriental looking man in Gray clothing, white tennis shoes, black well-groomed hair walking briskly.

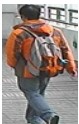 The male pedestrian is wearing jeans along with matching orange and Gray coat and backpack.

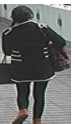 The young lady has a black bag on her left arm and is carrying a red and blue plaid garment on her right arm.

(a) CUHK-PEDES

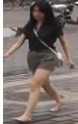 A slender woman with black hair is walking with a red umbrella over her head. She is wearing a black shirt with tan shorts and is carrying a white bag.

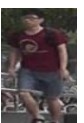 The man is wearing black and white tennis shoes with a pair of white socks. He is wearing a pair of grey jean shorts with a red tee-shirt with a white logo on the chest.

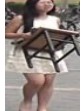 The lady has long black hair. She is wearing a sleeveless white dress that hangs to just above her knees. She is carrying a stool in her hands.

(b) ICFG-PEDES

Figure 6: Examples of CUHK-PEDES dataset and ICFG-PEDES dataset.

### B.2 More Experiment Details

To guarantee the performance of the baselines, we select the optimal temperature (Eq. 1) for the TTA baselines upon each dataset. According to Fig. 4(a), the temperature is fixed as 0.01 for COCO-C, Flickr-C, COCO, Flickr, and Nocaps datatsets, 0.001 for Fashion-Gen dataset, and 0.0001 for CUHK-PEDE and ICFG-PEDES datasets. Note that we employ Tent to conduct the experiment in Fig. 4(a) since most TTA methods are variants based on the Tent. Moreover, the adaptation process utilizes an initial learning rate of $3e^{-4}/3e^{-5}$ for text/image retrieval, excepting $3e^{-4}$ for image retrieval on the CLIP model.

In addition, for the ablation study in Fig. 4(a), we perform experiments on the Flickr-C dataset using the following corruptions: Gaussian, Zoom, Snow, and Contrast for the image modality; OCR, IP, and Formal for the text modality.

### B.3 More Details about the Related Work

Test-time Adaptation (TTA) aims to reconcile the distribution shifts in an online manner. To achieve this goal, fully TTA (Wang et al., 2021) has been proposed, which fine-tunes the BatchNorm layers by minimizing entropy during the test phase. EATA (Niu et al., 2022) employs a Fisher regularizer to limit excessive model parameter changes and filter out high-entropy samples via the selection strategy. SAR (Niu et al., 2023) removes high-gradient samples and promotes flat minimum weights, enhancing robustness against more challenging TTA scenarios such as mixed domain shifts, single-sample adaptation, and imbalanced label shifts. READ (Yang et al., 2024) proposes a noise-robust adaptation loss and reliable fusion module to tackle the reliability bias challenge in the multi-modal setting. DeYO (Lee et al., 2024) reveals the unreliability of treating entropy as the confidence metric and establishes a novel metric by measuring the difference between predictions before and after applying an object-destructive transformation.

### B.4 Pseudo Code

In the following, we provide the pseudo-code of the proposed TCR in Algorithm 1. To guarantee the stability of the estimation for $E_m$ and $\Delta_S$, we maintain a queue which always saves the query-candidate pairs with the smallest SI during the adaptation process. Following Caron et al. (2020), we limit the queue updating times to a maximum of 10 iterations.

---

**Algorithm 1: T**est-time adaptation for **C**ross-modal **R**etrieval (TCR)

**Input:** Test samples $\mathcal{D}_T = \left\{ \{\mathbf{x}_i^Q\}_{i=1}^{N^Q}, \{\mathbf{x}_j^G\}_{j=1}^{N^G} \right\}$, the source model $f_{\Theta_s}$ with trainable parameters $\tilde{\Theta}$,
      TTA steps $\eta > 0$, batch size $B$.
**Output:** Predictions $\{\mathbf{p}_i\}_{i=1}^{N^Q}$.

1  Initialize $\tilde{\Theta}_0 = \Theta_s$;
2  **for** *given queries* $\mathbf{x}^Q \in \mathcal{D}_T$ **do**
3     **for** *step* $= 1, \cdots, \eta$ **do**
4        Obtain the query predictions $\mathbf{p}$ in Eq. 1;
5        Select a subset of candidates $\mathbf{x}^{G'}$ from the gallery using Eq. 4 ;   // Candidate Selection
6        Obtain the refined query predictions $\hat{\mathbf{p}}$ in Eq. 5 and the corresponding entropy $E(\mathbf{x}^Q)$;
        // Update the queue
7        **if** *step* $= 1$ **then**
8           Compute the criterion SI in Eq. 6;
9           Select the 30% query-candidate pairs with the smallest $SI$;
10          Maintain a queue of size $B$ to save the pairs and their corresponding entropies;
11        **end**
12        Estimate the modality gap $\Delta_S$ using Eq. 7 ;          // Constraint Estimation
13        Estimate the desirable threshold $E_m$ using Eq. 8 ;      // Constraint Estimation
14        Compute the overall loss $\mathcal{L}$ in Eq. 3 with $\Delta_S$ and $E_m$;
15        Update parameters $\tilde{\Theta}$ through gradient descent to minimize $\mathcal{L}$;
16     **end**
17 **end**

---

## C  AN ALTERNATIVE IMPLEMENTATION OF TCR WITHOUT TRAINING

From the results in Fig. 3, we observe that the performance could be boosted by simply scaling up the uniformity or rectifying the modality gap even without adopting the function. Based on the observation, in this section, we propose to implement TCR in an untrained manner, which demonstrates the great potential of TCR. Specifically, to enhance the uniformity of the query modality, we scale the given queries by Eq. 12 with $\lambda^{\text{scale}}$ fixed at 2. To adjust the inter-modality gap, we estimate the modality gap $\Delta_S$ of the source domain by Eq. 7 and then rectify the modality gap in the target domain to $\Delta_S$ by Eq. 13. The details are presented in Algorithm 2 and the experiment results are shown in Table 10-11.

---

**Algorithm 2:** An Implementation of Untrained TCR

---

**Input:** Test samples $\mathcal{D}_T = \left\{ \{\mathbf{x}_i^Q\}_{i=1}^{N^Q}, \{\mathbf{x}_j^G\}_{j=1}^{N^G} \right\}$, the source model $f_{\Theta_s}$, batch size $B$, scaling factor
$\qquad \lambda^{\text{scale}}$.
**Output:** Predictions $\{\mathbf{p}_i\}_{i=1}^{N^Q}$.

1  Initialize $\tilde{\Theta}_0 = \Theta_s$;
2  **for** *given queries* $\mathbf{x}^Q \in \mathcal{D}_T$ **do**
3  $\quad$ Select a subset of candidates $\mathbf{x}^{G'}$ from the gallery using Eq. 4 ; $\qquad$ // Candidate Selection
$\quad$ // Update the queue
4  $\quad$ Compute the criterion SI in Eq. 6;
5  $\quad$ Select the 30% query-candidate pairs with the smallest $SI$;
6  $\quad$ Maintain a queue of size $B$ to save the pairs;
7  $\quad$ Scale $\mathbf{x}^Q$ using Eq. 12 with $\lambda^{\text{scale}}$ ; $\qquad$ // Scaling up Intra-modality Uniformity
8  $\quad$ Estimate the modality gap $\Delta_S$ using Eq. 7 ; $\qquad\qquad$ // Constraint Estimation
9  $\quad$ Rectify the modality gap to $\Delta_S$ using Eq. 13 ; $\qquad$ // Rectifying between-modality Gap
10 $\quad$ Perform $\ell 2$-normalization on the embeddings in the query modality;
11 $\quad$ Obtain the query predictions $\mathbf{p}$ in Eq. 1;
12 **end**

---

# D   MORE EXPERIMENT RESULTS

## D.1   RESULTS ON FLICKR-C

In the manuscript, we have carried out experiments on the COCO-C benchmark. Here, we provide more results on the Flickr-C benchmark. As shown in Table 7-8, TCR significantly outperforms all the baselines across various pre-trained model types and sizes on the Flickr-C benchmarks.

Table 7: Comparisons with state-of-the-art methods on Flickr-C benchmark under **QUERY SHIFT ON THE IMAGE MODALITY** with maximum severity level regarding the Recall@1 metric.

| | Noise | | | | Blur | | | | Weather | | | | Digital | | | | |
|---|---|---|---|---|---|---|---|---|---|---|---|---|---|---|---|---|---|
| Query Shift | Gauss. | Shot | Impul. | Speckle | Defoc. | Glass | Motion | Zoom | Snow | Frost | Fog | Brit. | Contr. | Elastic | Pixel | JPEG | Avg. |
| BLIP ViT-B/16 | 49.8 | 56.6 | 50.3 | 71.6 | 53.1 | 84.5 | 47.4 | 15.5 | 66.4 | 80.4 | 79.5 | 85.5 | 60.6 | 53.3 | 35.1 | 80.3 | 60.6 |
| • Tent | 54.9 | 54.9 | 54.3 | 73.1 | 53.3 | 85.3 | 47.9 | 1.6 | 67.2 | 80.9 | 79.6 | 86.8 | 63.6 | 53.4 | 35.4 | 81.4 | 60.9 |
| • EATA | 55.5 | 60.5 | 55.8 | 75.8 | 64.6 | 86.2 | 52.2 | 8.5 | 72.0 | 83.7 | 82.5 | 87.9 | 68.4 | 60.1 | 45.9 | 81.6 | 65.1 |
| • SAR | 54.8 | 62.5 | 55.6 | 75.2 | 48.3 | 87.2 | 34.8 | 15.5 | 71.9 | 83.1 | 82.2 | 87.9 | 68.2 | 60.3 | 42.2 | 81.4 | 63.2 |
| • READ | 50.1 | 58.2 | 52.2 | 74.8 | 63.7 | 87.0 | 55.1 | 2.2 | 71.7 | 83.8 | 81.9 | 87.7 | 67.4 | 62.3 | 42.5 | 81.4 | 63.9 |
| • DeYO | 55.4 | 62.0 | 56.3 | 76.2 | 63.8 | 86.3 | 50.3 | 3.2 | 73.1 | 84.1 | 83.2 | 88.6 | 70.1 | 63.1 | 46.8 | 81.3 | 65.2 |
| • Ours | **62.0** | **66.6** | **61.4** | **80.0** | **68.1** | **87.9** | **65.2** | **39.9** | **78.2** | **85.2** | **85.7** | **89.5** | **75.1** | **73.1** | **56.8** | **83.3** | **72.4** |
| BLIP ViT-L/16 | 58.2 | 61.0 | 59.7 | 76.9 | 66.4 | 88.5 | 62.5 | 33.4 | 67.7 | 81.5 | 79.3 | 89.1 | 60.4 | 66.4 | 46.5 | 85.0 | 67.7 |
| • Tent | 61.3 | 64.3 | 63.3 | 77.6 | 70.8 | 88.7 | 62.8 | 31.5 | 70.4 | 83.8 | 81.1 | 89.2 | 61.2 | 68.7 | 52.0 | 84.5 | 69.5 |
| • EATA | 62.0 | 65.1 | 64.5 | 78.9 | 70.2 | 89.5 | 63.3 | 33.1 | 71.9 | 83.7 | 81.2 | 89.3 | 61.6 | 69.3 | 53.0 | 85.3 | 70.2 |
| • SAR | 61.1 | 64.4 | 63.7 | 79.7 | 71.6 | 90.3 | 64.4 | 27.6 | 70.6 | 83.4 | 81.0 | 89.7 | 62.4 | 70.1 | 53.3 | 85.3 | 69.9 |
| • READ | 61.1 | 64.4 | 63.7 | 79.7 | 71.6 | 90.3 | 64.4 | 27.6 | 70.6 | 83.4 | 81.0 | 89.7 | 62.4 | 70.1 | 53.3 | 85.3 | 69.9 |
| • DeYO | 61.5 | 61.0 | 62.1 | 78.3 | 69.6 | 89.5 | 62.5 | 37.2 | 72.1 | 83.6 | 81.4 | 89.9 | 61.3 | 67.6 | 52.5 | 86.8 | 69.8 |
| • Ours | **68.2** | **71.7** | **70.2** | **83.3** | **74.7** | **91.9** | **72.5** | **49.6** | **78.2** | **87.0** | **85.5** | **92.1** | **70.9** | **79.6** | **65.5** | **87.8** | **76.8** |

Table 8: Comparisons with state-of-the-art methods on Flickr-C benchmark under **QUERY SHIFT ON THE TEXT MODALITY** with maximum severity level regarding the Recall@1 metric.

| | Character-level | | | | | Word-level | | | | | Sentence-level | | | | | |
|---|---|---|---|---|---|---|---|---|---|---|---|---|---|---|---|---|
| Query Shift | OCR | CI | CR | CS | CD | SR | RI | RS | RD | IP | Formal | Casual | Passive | Active | Backtrans | Avg. |
| BLIP ViT-B/16 | 53.5 | 18.4 | 18.0 | 30.4 | 22.5 | 68.3 | 77.9 | 76.9 | 77.9 | 82.1 | 82.1 | 81.9 | 79.9 | 82.2 | 79.8 | 62.1 |
| • Tent | 55.4 | 18.6 | 18.2 | 31.1 | 23.0 | 69.6 | 78.8 | 77.7 | 78.0 | 82.2 | 81.9 | 81.8 | 79.6 | 82.0 | 79.9 | 62.5 |
| • EATA | 55.7 | 19.9 | 19.9 | 31.6 | 23.6 | 69.5 | 78.6 | 77.5 | 77.9 | 82.4 | 82.3 | 81.8 | 80.5 | **82.6** | 80.2 | 62.9 |
| • SAR | 53.5 | 20.1 | 19.1 | 32.1 | 23.8 | 68.3 | 77.9 | 76.9 | 77.9 | 82.1 | 82.1 | 81.9 | 79.9 | 82.2 | 79.8 | 62.5 |
| • READ | 55.8 | 19.7 | 20.6 | 32.0 | 23.5 | 69.3 | 78.6 | 77.6 | 78.1 | 82.4 | 82.2 | 81.8 | 80.5 | 82.5 | 80.2 | 63.0 |
| • DeYO | 53.5 | 18.4 | 18.0 | 30.4 | 22.5 | 68.3 | 77.9 | 76.9 | 77.9 | 82.1 | 82.1 | 81.9 | 79.9 | 82.2 | 79.8 | 62.1 |
| • Ours | **57.1** | **21.4** | **22.5** | **33.6** | **25.1** | **69.8** | **79.3** | **78.0** | **78.1** | **82.5** | **82.4** | **82.2** | **81.0** | **82.6** | 80.2 | **63.7** |
| BLIP ViT-L/16 | 58.0 | 22.2 | 22.0 | 34.1 | 25.1 | 71.2 | 79.9 | 78.9 | 78.8 | 83.3 | 83.1 | 82.7 | 81.7 | 83.5 | 80.7 | 64.4 |
| • Tent | 59.0 | 22.4 | 22.1 | 34.5 | 25.3 | 71.4 | 80.3 | 79.3 | 78.8 | 83.7 | 82.8 | 82.7 | 81.8 | 83.3 | 80.7 | 64.6 |
| • EATA | 59.1 | 23.0 | 23.2 | 35.1 | 25.6 | 71.7 | 80.3 | 79.3 | 78.8 | 83.5 | 83.0 | 83.2 | 81.8 | **83.5** | 80.7 | 64.8 |
| • SAR | 58.1 | 23.1 | 23.0 | 34.5 | 25.8 | 71.2 | 79.9 | 78.9 | 78.8 | 83.3 | 83.1 | 82.7 | 81.7 | 83.4 | 80.7 | 64.6 |
| • READ | 58.9 | 23.4 | 23.3 | 34.9 | 25.9 | 71.5 | 80.7 | 79.3 | 78.8 | 83.5 | **83.2** | 83.1 | **81.9** | 83.4 | **80.8** | 64.8 |
| • DeYO | 58.1 | 22.2 | 22.0 | 34.1 | 25.1 | 71.2 | 79.9 | 78.9 | 78.7 | 83.3 | 83.1 | 82.7 | 81.7 | 83.4 | **80.8** | 64.4 |
| • Ours | **59.7** | **24.4** | **24.4** | **36.1** | **26.7** | **71.8** | **80.9** | **79.5** | **78.9** | **83.5** | **83.2** | **83.4** | 81.8 | **83.5** | **80.8** | **65.2** |

## D.2   EXPERIMENTS ABOUT PERSONALIZED QUERIES ON FASHION-GEN

As mentioned in Introduction of the manuscript, different inquirers would submit personalized queries, *e.g.*, some are drawn to fashionable handbags, while others are passionate about collecting a variety of shoes. To further demonstrate the generalization of TCR in this scenario, we simulate the personalized queries on the Fashion-Gen benchmark. In detail, following Cartella et al. (2023), we fine-tune the pre-trained CLIP on four publicly available fashion datasets including Fashion-Gen, Fashion IQ (Wu et al., 2021), Fashion200K (Han et al., 2017), and iMaterialist (Guo et al., 2019). After that, we employ TCR to adapt various preferences such as "TOPS" and "SWEATERS" on the Fashion-Gen benchmark. The experimental results are presented in Table 9, one could observe that TCR improves both image-to-text and text-to-image retrieval performance under personalized queries.

Table 9: The cross-modal retrieval performance of TCR on Fashion-Gen benchmark with **PERSONALIZED QUERIES** regarding Recall@1 metric.

| | Query Shift | TOPS | SWEATERS | JACKETS | PANTS | JEANS | SHIRTS | DRESSES | SHORTS | SNEAKERS | SKIRTS | Avg. |
|---|---|---|---|---|---|---|---|---|---|---|---|---|
| TR | CLIP ViT-B/32 | 18.0 | 19.3 | 19.9 | 12.0 | 5.5 | 18.3 | 38.1 | **17.9** | 37.3 | 29.6 | 21.6 |
| | • Ours | **22.9** | **25.2** | **21.6** | **14.3** | **6.0** | **22.8** | **44.3** | 8.5 | **41.7** | **37.4** | **24.5** |
| IR | CLIP ViT-B/32 | 24.9 | 27.9 | 29.2 | 16.9 | 6.7 | 25.4 | 51.8 | 25.7 | 47.1 | 47.8 | 30.3 |
| | • Ours | **28.2** | **31.7** | **32.8** | **19.5** | **9.6** | **28.5** | **57.1** | **29.1** | **53.6** | **50.7** | **34.1** |

## D.3   Results on the All Severity Setting

In Section 4.2 of the manuscript and Appendix D.1, we have demonstrated the effectiveness of TCR in handling query shift at the maximum severity level. To further verify the robustness of TCR, we conduct more experiments on the COCO-C benchmark with query shift across all severity levels.

The results in Fig. 7-8 indicate the effectiveness of TCR in addressing various severity of the query shift.

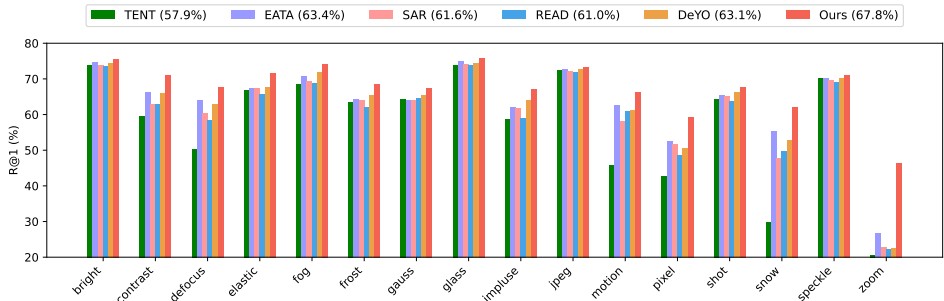

Figure 7: Text retrieval performance comparisons on the COCO-C benchmark under **QUERY SHIFT ON THE IMAGE MODALITY** with all severity levels regarding Recall@1 metric. The legend key provides an overview of the average performance of each approach across various corruption types.

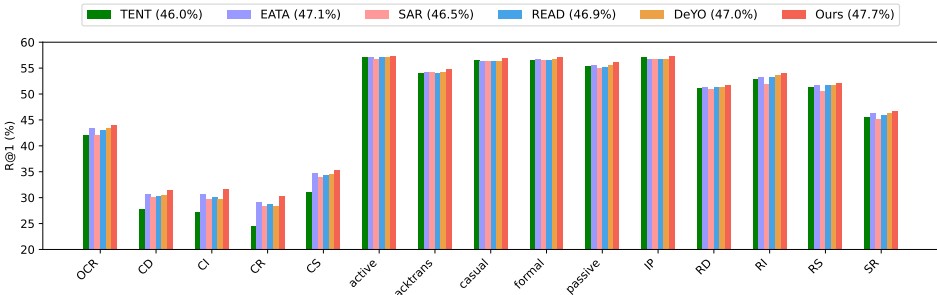

Figure 8: Image retrieval performance comparisons on the COCO-C benchmark under **QUERY SHIFT ON THE TEXT MODALITY** with all severity levels regarding Recall@1 metric.

### D.4 RESULTS OF UNTRAINED TCR

Here, we perform experiments on the COCO-C and Flickr-C benchmarks to evaluate the proposed untrained TCR in Appendix C. During the experiments, we compare untrained TCR with the best baseline EATA in Table 1-2. From the results in Table 10 and Table 11, we observe that the untrained TCR achieves significant improvement over EATA, even without parameter update, which corroborates our observations and validates the effectiveness of the proposed TCR.

Table 10: The cross-modal retrieval performance of untrained TCR on COCO-C and Flickr-C benchmarks under **IMAGE MODALITY DISTRIBUTION SHIFTS** with maximum severity level regarding the Recall@1 metric.

| Dataset | Query Shift | Noise | | | | Blur | | | | Weather | | | | Digital | | | | Avg. |
|---|---|---|---|---|---|---|---|---|---|---|---|---|---|---|---|---|---|---|
| | | Gauss. | Shot | Impul. | Speckle | Defoc. | Glass | Motion | Zoom | Snow | Frost | Fog | Brit. | Contr. | Elastic | Pixel | JPEG | |
| Flickr-C | EATA | 55.5 | 60.5 | 55.8 | 75.8 | 64.6 | 86.2 | 52.2 | 8.5 | 72.0 | 83.7 | 82.5 | 87.9 | 68.4 | 60.1 | 45.9 | 81.6 | 65.1 |
| | Ours (untrain) | 58.7 | 63.2 | 58.1 | 78.8 | 65.9 | 87.8 | 61.2 | 34.6 | 79.2 | 84.8 | 84.4 | 89.1 | 68.2 | 67.4 | 46.0 | 83.0 | 69.4 |
| | Ours | **62.0** | **66.6** | **61.4** | **80.0** | **68.1** | **87.9** | **65.2** | **39.9** | **78.2** | **85.2** | **85.7** | **89.5** | **75.1** | **73.1** | **56.8** | **83.3** | **72.4** |
| COCO-C | EATA | 41.4 | 50.3 | 35.7 | 63.1 | 49.8 | 72.2 | 46.2 | 6.9 | 45.6 | 56.7 | 62.5 | 71.4 | 43.6 | 51.3 | 25.6 | 67.0 | 49.3 |
| | Ours (untrain) | 48.8 | 51.7 | 49.8 | 61.5 | 53.9 | 72.6 | 49.4 | 18.7 | 49.7 | 60.5 | 67.1 | 71.4 | 43.9 | 49.9 | 26.7 | 67.4 | 52.7 |
| | Ours | **53.2** | **56.2** | **54.8** | **64.6** | **58.0** | **73.7** | **56.4** | **32.2** | **56.5** | **64.1** | **71.0** | **73.4** | **57.9** | **63.7** | **41.8** | **68.4** | **59.1** |

Table 11: The cross-modal retrieval performance of untrained TCR on COCO-C and Flickr-C benchmarks under **TEXT MODALITY DISTRIBUTION SHIFTS** with maximum severity level regarding the Recall@1 metric.

| Dataset | Query Shift | Character-level | | | | | Word-level | | | | | Sentence-level | | | | | Avg. |
|---|---|---|---|---|---|---|---|---|---|---|---|---|---|---|---|---|---|
| | | OCR | CI | CR | CS | CD | SR | RI | RS | RD | IP | Formal | Casual | Passive | Active | Backtrans | |
| Flickr-C | EATA | 55.7 | 19.9 | 19.9 | 31.6 | 23.6 | 69.5 | 78.6 | 77.5 | 77.9 | 82.4 | 82.3 | 81.8 | 80.5 | **82.6** | 80.2 | 62.9 |
| | Ours (untrain) | 55.8 | 20.3 | 20.7 | 32.7 | 23.8 | 69.2 | 78.3 | 77.8 | 77.8 | 82.5 | 82.2 | 82.0 | 80.4 | 82.3 | 80.0 | 63.1 |
| | Ours | **57.1** | **21.4** | **22.5** | **33.6** | **25.1** | **69.8** | **79.3** | **78.0** | **78.1** | **82.5** | **82.4** | **82.2** | **81.0** | 82.6 | 80.2 | **63.7** |
| COCO-C | EATA | 33.1 | 11.9 | 10.5 | 18.4 | 12.0 | 44.9 | 53.0 | 51.6 | 50.3 | 56.2 | 56.8 | 56.8 | 56.0 | 56.8 | 54.3 | 41.5 |
| | Ours (untrain) | 32.9 | 12.3 | 10.4 | 19.0 | 12.3 | 44.8 | 52.6 | 51.3 | 51.5 | 57.8 | 57.1 | **57.2** | **56.2** | 57.2 | **54.7** | 41.8 |
| | Ours | **34.1** | **13.7** | **11.8** | **19.5** | **13.2** | **45.3** | **53.8** | **51.8** | **51.5** | **57.3** | **57.1** | 56.8 | 56.0 | **57.3** | **54.7** | **42.3** |

## D.5 MORE ABLATION RESULTS ABOUT MODALITY UNIFORMITY AND MODALITY GAP

We provide more ablation studies under the "Base2COCO" setting to prove that TCR achieves better performance by enlarging intra-modality uniformity and rectifying the inter-modality gap. Specifically, we present the intra-modality uniformity and inter-modality gap of different baselines after TTA in Table 12. The results illustrate that i) most of the baselines improve the performance by implicitly enlarging the intra-modality uniformity and narrowing the modality gap; ii) the improvement of Tent is unstable due to the enlarged modality gap; iii) the proposed TCR achieves the highest intra-modality uniformity (0.93 and 0.96) and enjoys the modality gap (0.63 and 0.64) in the target domain close to that (0.67) in the source domain, thus contributing to boosting the performance. Notably, we obtain the modality gap in the source domain by constructing a subset of 12,000 image-text pairs derived from the COCO, Visual Genome (Krishna et al., 2017), CC3M (Changpinyo et al., 2021), and SBU Captions (Ordonez et al., 2011) datasets.

Table 12: The intra-modality uniformity and inter-modality gap of different baselines after TTA under the "Base2COCO" setting. IU and TU indicate the uniformity of image and text modalities, respectively. MG indicates the modality gap.

| Method | IU | MG | TR@1 | TU | MG | IR@1 |
|--------|------|------|------|------|------|------|
| Base | 0.62 | 0.72 | 59.3 | 0.67 | 0.72 | 45.4 |
| Tent | 0.82 | 0.74 | 61.7 | 0.85 | 0.76 | 41.7 |
| EATA | 0.87 | 0.68 | 64.2 | 0.88 | 0.67 | 47.9 |
| SAR | 0.86 | 0.70 | 63.5 | 0.74 | 0.69 | 46.6 |
| READ | 0.85 | 0.72 | 62.1 | 0.84 | 0.70 | 46.4 |
| DeYO | 0.88 | 0.68 | 65.0 | 0.86 | 0.67 | 47.3 |
| Ours | 0.93 | 0.63 | 68.9 | 0.96 | 0.64 | 48.9 |

## D.6 MORE VISUALIZATION RESULT

As shown in Fig. 4(c) of the manuscript, we have visualized the text retrieval results before/after TTA. Here, we provide additional visualization results of image retrieval before/after TTA under the "Base2COCO" setting. The results in Fig. 9 illustrate that samples in the query modality enjoy more scatter and the modality gap narrows after the TTA process, which proves that TCR improves performance in both TR and IR by rectifying the intra-modality uniformity and the inter-modality gap.

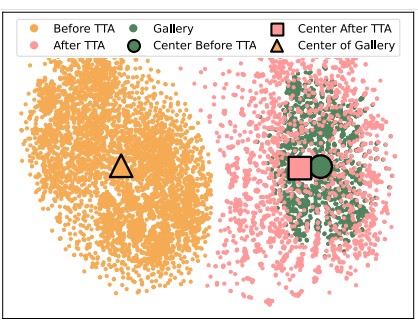

Figure 9: The t-SNE visualization results of image retrieval on the query and gallery embeddings by employing the proposed TCR.

### D.7 Results on the Mixed Severity Setting and Mixed Query Shift

We conduct more experiments on the COCO-C benchmark, investigating the robustness of the proposed TCR under non-i.i.d. settings (i.e., Mixed Severity Levels and Mixed Corruption Types). Specifically,

- Mixed Severity Levels: For each corruption, we create the test pairs by selecting $1/m$ of the data from each severity level, resulting in a total of $N$ test pairs, where $m$ is the number of severity levels and m=5 / 7 / 2 for the image / character-level / word-level corruptions.
- Mixed Corruption Types: For the text retrieval, we construct the test pairs by selecting 1/16 of the data from each image corruption (1 through 16), resulting in a total of $N$ test pairs. For the image retrieval, we create test pairs by selecting 1/15 of the data from each text corruption (1 through 15), resulting in a total of $N$ test pairs.

To verify the effectiveness of TCR under the Mixed Severity and Mixed Corruption Types settings, we choose the typical TTA method Tent and the SOTA TTA methods EATA, DeYO as baselines for comparisons. In the experiment, we carry out the experiments on the COCO-C benchmark, and he corresponding results are depicted in Table 13-15. Note that for the Mixed Corruption Types setting, there are five levels of the mixed corruptions in text retrieval, corresponding to the image corruptions with five severity levels. For image retrieval, the severity levels of character-level/word-level/sentence-level text corruptions are 7/2/1. Thus, we select the two highest severity levels for character-level and word-level corruptions, and combine them with sentence-level corruptions, resulting in two levels of the mixed corruptions.

The performance superiority of TCR over all baselines under both Mixed Severity and Mixed Corruption Types settings demonstrates its robustness against non-i.i.d query shift.

Table 13: Comparisons with state-of-the-art methods on COCO-C benchmark under **Mixed Severity Levels on the image modality** regarding the Recall@1 metric.

| Mixed Severity Levels | Noise | | | Speckle | Blur | | | | Weather | | | | Digital | | | | Avg. |
|---|---|---|---|---|---|---|---|---|---|---|---|---|---|---|---|---|---|
| | Gauss. | Shot | Impul. | | Defoc. | Glass | Motion | Zoom | Snow | Frost | Fog | Brit. | Contr. | Elastic | Pixel | JPEG | |
| BLIP ViT-B/16 | 61.8 | 61.8 | 59.7 | 66.4 | 58.5 | 70.7 | 56.7 | 22.5 | 42.6 | 60.4 | 66.0 | 70.8 | 61.0 | 61.2 | 47.8 | 69.8 | 58.6 |
| • Tent | 65.3 | 64.9 | 59.9 | 69.4 | 31.6 | 74.1 | 35.7 | 1.9 | 10.7 | 63.3 | 70.4 | 73.8 | 64.4 | 65.8 | 47.8 | 71.5 | 54.4 |
| • EATA | 64.9 | 65.6 | 64.6 | 70.0 | 62.0 | 74.3 | 61.7 | 28.1 | 55.3 | 63.9 | 71.1 | 74.4 | 65.5 | 66.0 | 53.7 | 72.7 | 63.4 |
| • DeYO | 64.0 | 66.0 | 63.0 | 69.8 | 64.6 | 74.6 | 63.0 | 5.8 | 56.1 | 65.7 | 71.4 | 74.5 | 65.7 | 67.8 | 52.5 | 72.7 | 62.3 |
| • Ours | **67.2** | **68.1** | **66.6** | **70.7** | **67.0** | **75.8** | **65.8** | **45.7** | **61.2** | **68.4** | **74.2** | **75.2** | **70.4** | **70.4** | **58.6** | **73.5** | **67.4** |

Table 14: Comparisons with state-of-the-art methods on COCO-C benchmark under **Mixed Severity Levels on the text modality** regarding the Recall@1 metric.

| Mixed Severity Levels | Character-level | | | | | Word-level | | | | | Avg. |
|---|---|---|---|---|---|---|---|---|---|---|---|
| | OCR | CI | CR | CS | CD | SR | RI | RS | RD | IP | |
| BLIP ViT-B/16 | 42.1 | 29.7 | 28.0 | 33.9 | 30.0 | 44.8 | 51.7 | 50.5 | 50.8 | 56.8 | 41.8 |
| • Tent | 42.4 | 28.5 | 23.6 | 33.8 | 26.9 | 45.5 | 52.4 | 51.4 | 50.9 | 57.0 | 41.2 |
| • EATA | 43.4 | 30.8 | 29.4 | 34.8 | 30.7 | 46.0 | 53.2 | 51.8 | 51.4 | 57.6 | 42.9 |
| • DeYO | 43.4 | 30.7 | 29.3 | 35.0 | 30.9 | 46.2 | 53.4 | 51.9 | 51.4 | **57.7** | 43.0 |
| • Ours | **44.4** | **32.2** | **30.6** | **35.7** | **31.7** | **46.3** | **53.8** | **52.1** | **51.5** | 57.4 | **43.6** |

Table 15: Comparisons with state-of-the-art methods on COCO-C benchmark under **Mixed Corruption Types** levels regarding the Recall@1 metric.

| Mixed Corruption Types | TR@1 | | | | | IR@1 | | Avg. |
|---|---|---|---|---|---|---|---|---|
| | Level 1 | Level 2 | Level 3 | Level 4 | Level 5 | Level 1 | Level 2 | |
| BLIP ViT-B/16 | 68.8 | 64.5 | 61.4 | 54.5 | 44.9 | 42.5 | 41.1 | 53.9 |
| • Tent | 70.0 | 67.0 | 64.4 | 56.4 | 33.2 | 42.2 | 38.5 | 53.1 |
| • EATA | 71.7 | 68.2 | 64.7 | 58.9 | 48.0 | 43.2 | 41.9 | 56.7 |
| • DeYO | 71.5 | 68.4 | 65.1 | 59.9 | 48.3 | 42.9 | 41.8 | 56.8 |
| • Ours | **73.3** | **70.4** | **66.9** | **61.5** | **53.6** | **43.8** | **42.3** | **58.8** |

D.8 RESULTS ON THE GALLERY SHIFT

In the manuscript, we have conducted experiments under the Query-Gallery Shift setting (Table 3), which demonstrates that TCR could improve retrieval performance even when the gallery modality occurs distribution shift. Here, we conduct additional experiments to investigate whether gallery shift would affect the outcomes of nearest neighbor selection. Specifically, we carry out experiments on the COCO-C benchmark under two gallery shift settings, i.e., only gallery shift setting, both query and gallery shift setting.

- Only Gallery Shift: In this setting, there is no distribution shift in the query modality. For the baseline BLIP ViT-B/16, the IR@1 and TR@1 without any query or gallery shift are 57.1% and 74.0%, respectively.
- Both Query and Gallery Shift: In this setting, we choose the OCR/Gaussian corruptions as the query shift for image/text retrieval, respectively. For the baseline BLIP ViT-B/16, the IR@1/TR@1 with OCR/Gaussian corruptions is 31.4%/43.4%.

From the results in Table 16-17, one could observe that gallery shift degrades both retrieval performance and nearest neighbor selection accuracy, whether in only gallery shift or both query and gallery shift settings. However, the proposed TCR improves the retrieval performance under gallery shift, with the selected nearest neighbors more likely to be correct. Besides, even under gallery shift setting, TCR could enhance retrieval performance surpassing the baseline performance without gallery shift. For example, in the both query and gallery shift setting, the text retrieval performance of TCR under RI (47.7%), RS (45.4%), Formal (52.4%), and Passive (52.1%) gallery shift exceeds the baseline performance without gallery shift (43.4%). It's worth noting that in real-world scenarios, data with only gallery shift is rare, as the data in the gallery is often extensive and curated. In contrast, the queries of the users are more diverse, which might lead to the distribution shift challenge.

Table 16: Performance under **GALLERY SHIFT ON THE IMAGE MODALITY** regarding the Recall@1 metric and neighbor ACC (i.e., the cross-modal nearest neighbor of the query is correct). Notably, for the baseline BLIP ViT-B/16, the neighbor ACC and R@1 are the same since both are computed using cosine similarity for ranking.

| Image Retrieval | Noise Gauss. | Shot | Impul. | Speckle | Blur Defoc. | Glass | Motion | Zoom | Weather Snow | Frost | Fog | Brit. | Digital Contr. | Elastic | Pixel | JPEG | Avg. |
|---|---|---|---|---|---|---|---|---|---|---|---|---|---|---|---|---|---|
| No Query Shift (ACC & R@1) | 35.4 | 37.4 | 35.9 | 44.4 | 38.5 | 53.2 | 35.8 | 15.2 | 36.5 | 43.0 | 47.7 | 51.7 | 32.5 | 36.2 | 20.3 | 48.9 | 38.3 |
| • TCR (ACC) | 36.3 | 38.0 | 36.6 | 45.2 | 39.1 | 53.7 | 37.4 | 16.4 | 38.4 | 44.2 | 49.2 | 52.5 | 33.1 | 38.5 | 21.8 | 49.5 | 39.4 |
| • TCR (R@1) | 36.5 | 38.6 | 37.1 | 45.2 | 39.9 | 53.8 | 37.4 | 16.8 | 38.3 | 44.5 | 49.3 | 52.4 | 33.5 | 38.5 | 21.8 | 49.6 | 39.6 |
| OCR Corruption (ACC & R@1) | 18.5 | 19.8 | 18.6 | 23.4 | 20.1 | 28.8 | 19.0 | 8.2 | 18.9 | 22.5 | 25.5 | 27.6 | 17.1 | 18.6 | 10.4 | 26.0 | 20.2 |
| • TCR (ACC) | 20.6 | 21.8 | 20.6 | 26.0 | 22.6 | 31.6 | 21.5 | 9.3 | 21.6 | 25.5 | 28.8 | 30.5 | 18.9 | 21.8 | 12.1 | 28.4 | 22.6 |
| • TCR (R@1) | 20.6 | 21.8 | 20.6 | 26.0 | 22.6 | 31.7 | 21.5 | 9.4 | 21.6 | 25.6 | 28.8 | 30.5 | 19.0 | 21.8 | 12.2 | 28.4 | 22.6 |

Table 17: Performance under **GALLERY SHIFT ON THE TEXT MODALITY** regarding the Recall@1 metric and neighbor ACC.

| Text Retrieval | Character-level OCR | CI | CR | CS | CD | Word-level SR | RI | RS | RD | IP | Setence-level Formal | Casual | Passive | Active | Backtrans | Avg. |
|---|---|---|---|---|---|---|---|---|---|---|---|---|---|---|---|---|
| No Query Shift (ACC & R@1) | 49.7 | 23.1 | 20.1 | 34.5 | 22.8 | 59.7 | 64.8 | 65.2 | 66.9 | 73.1 | 73.2 | 72.5 | 71.5 | 73.6 | 71.1 | 56.1 |
| • TCR (ACC) | 55.5 | 26.8 | 23.5 | 39.5 | 27.5 | 66.2 | 70.9 | 70.4 | 71.3 | 77.1 | 76.5 | 76.2 | 75.1 | 77.3 | 75.4 | 60.6 |
| • TCR (R@1) | 55.5 | 26.8 | 23.5 | 39.0 | 27.7 | 65.9 | 70.9 | 70.6 | 71.6 | 77.2 | 76.7 | 76.2 | 75.2 | 77.3 | 75.3 | 60.6 |
| Gauss. Corruption (ACC & R@1) | 27.2 | 13.2 | 11.1 | 18.4 | 27.2 | 32.5 | 36.3 | 37.2 | 38.9 | 42.2 | 42.8 | 42.7 | 41.0 | 43.2 | 41.0 | 33.0 |
| • TCR (ACC) | 35.7 | 16.0 | 15.2 | 24.7 | 35.7 | 42.1 | 47.1 | 46.0 | 48.2 | 53.0 | 53.8 | 53.0 | 51.9 | 53.1 | 51.7 | 41.8 |
| • TCR (R@1) | 35.2 | 16.5 | 15.3 | 24.6 | 35.2 | 42.2 | 47.7 | 45.4 | 48.3 | 52.8 | 52.4 | 52.7 | 52.1 | 53.5 | 52.1 | 41.7 |

### D.9 MORE ABLATION RESULTS ABOUT QUERY REFINEMENT MODULE

We conduct more experiments to investigate how the number of selected candidates affects performance. To this end, we directly perform zero-shot retrieval experiment on the COCO dataset with pre-trained BLIP as the source model. In the paper, we retrieve the most similar sample from the gallery set for each query, thus the number of selected candidates is equal to the batch size $B$. In the additional experiment, we vary the number of selected candidates at different values, i.e., $[0.2B, 0.5B, B, 2B, 5B, 10B, 50B]$. Specifically, assume that the number of selected candidates is $\lambda B$, where $\lambda$ is an integer. When $\lambda < 1$, we randomly select $\lambda B$ candidates from the original $B$ selected candidates. When $\lambda \geq 1$, we retrieve the most similar $\lambda$ candidates from the gallery set for each query, forming a new set of $\lambda B$ selected candidates.

From the results in Table 18, one could observe that increasing the number of selected candidates would significantly degrade the performance. Such a phenomenon indicates that an excessively large number of candidates may lead to underfitting issue, which highlights the necessity and effectiveness of the query refinement module.

Table 18: Ablation study on the number of selected candidates under "Base2COCO" setting.

| Number | $0.2B$ | $0.5B$ | $B$ | $2B$ | $5B$ | $10B$ | $50B$ |
|---|---|---|---|---|---|---|---|
| TR@1 | 67.2 | 68.5 | 68.9 | 65.3 | 64.9 | 64.7 | 64.6 |
| IR@1 | 47.3 | 48.3 | 48.9 | 48.3 | 48.2 | 48.0 | 47.5 |

### D.10 EFFICIENCY COMPARISONS

In this section, we conduct additional experiments to analyze the efficiency of TCR. To this end, we choose the pre-trained model BLIP as the source model and perform zero-shot retrieval on the COCO dataset. We measure the GPU time during the test-time adaptation phase. Note that the learnable parameters of all the methods are the same for a fair comparison. The results underscore that TCR achieves adaptation more efficiently than the augmentation-based method DeYO. Compared to the vanilla Tent and EATA (only low-entropy samples are employed for optimization), TCR requires only a negligible additional time cost, primarily due to the nearest neighbor selection in the query prediction refinement module.

Table 19: Efficiency comparisons among different approaches under "Base2COCO" setting.

| Method | TR | IR | Avg. |
|---|---|---|---|
| Tent | 285.5 seconds | 189.7 seconds | 237.6 seconds |
| EATA | 276.3 seconds | 190.4 seconds | 233.3 seconds |
| DeYO | 391.6 seconds | 254.2 seconds | 322.9 seconds |
| Ours | 291.1 seconds | 193.6 seconds | 242.4 seconds |

## D.11 RESULTS IN REMOTE SENSING DOMAIN

We conduct additional experiments in the even rarer remote sensing domain. To this end, we choose the BLIP as the source model and perform zero-shot retrieval on the remote sensing datasets RSICD (Lu et al., 2017) and RSITMD (Yuan et al., 2022). To verify the effectiveness of TCR, we choose the typical TTA method Tent, the SOTA TTA method EATA and DeYO as the baselines for comparisons. The results in Table 20 indicate that TCR could also achieve the best performance in even rarer remote sensing domain.

Table 20: Comparisons with state-of-the-art methods on benchmarks in the remote sensing domain with **QUERY-GALLERY SHIFT** regarding the Recall@1 metric.

| Query Shift | Base2RSICD | | Base2RSITMD | | |
| --- | --- | --- | --- | --- | --- |
| | TR@1 | IR@1 | TR@1 | IR@1 | Avg. |
| BLIP ViT-B/16 | 6.4 | 6.8 | 7.6 | 10.4 | 7.8 |
| • Tent | 5.7 | 5.4 | 7.9 | 9.3 | 7.1 |
| • EATA | 6.9 | 6.7 | 8.0 | 10.4 | 8.0 |
| • DeYO | 6.4 | 6.5 | 7.7 | 10.0 | 7.7 |
| • Ours | **8.5** | **7.1** | **8.4** | **10.7** | **8.7** |

## E MORE DISCUSSIONS ABOUT TEMPORAL SHIFT AND CONCEPT DRIFT

In this section, we discuss the connection between query shift, temporal shift, and concept drift. To be specific, as discussed in (Yu et al., 2024), the underlying distributions of data are distinct at different times, leading to *concept drift*. For instance, in weather forecasting, data is collected across diverse distributions (e.g., sunny, frost, snow). It is noteworthy that we have evaluated TCR under query shift caused by different weather conditions (e.g., frost, snow, fog). The corresponding results from COCO-C (Table 1) and Flickr-C (Table 7) settings demonstrate the robustness of TCR against concept drift to some extent. As noted in (Xie et al., 2024) and (Bai et al., 2022), the collected data would continuously vary over time, resulting in *temporal shift*. For example, changes in lighting conditions throughout the day could impact the distribution of the collected data. For evaluation, we have conducted experiments under the CUHK2ICFG setting (Table 4). Specifically, the ICFG-PEDES dataset is gathered at different times of the day (i.e., morning, noon, and afternoon), while the CUHK-PEDES dataset is derived from short-duration surveillance videos. Therefore, compared to CUHK-PEDES, the data in the ICFG-PEDES dataset exhibit distribution shifts due to time changes, such as illumination variation. The corresponding results from the CUHK2ICFG setting demonstrate that TCR could achieve robustness against temporal shift to some extent. Notably, any distribution shifts in the query modality would lead to query shift, not limited to temporal issues.

