# OpenReview forum: "Test-time Adaptation for Cross-modal Retrieval with Query Shift"
_ICLR.cc/2025/Conference — ICLR 2025 Spotlight_

### Official Review · Reviewer_jDwD · 2024-10-28

**Soundness:** 3
**Presentation:** 3
**Contribution:** 2
**Rating:** 8
**Confidence:** 4

**Summary:**

This paper introduces a novel method named TCR for addressing the query shift problem in cross-modal retrieval. TCR employs a test-time adaptation approach that leverages a multi-scale adaptive convolutional neural network and a hybrid transformer module to refine query predictions and adapt to shifts in query distribution without additional training data. The method is designed to enhance the uniformity of the query modality and reduce the gap between query and gallery modalities, thereby improving retrieval performance. The study demonstrates TCR's effectiveness on image-text retrieval tasks using standard benchmarks and various corruption types.

**Strengths:**

1.	The paper proposes a novel test-time adaptation method (TCR) to address the query shift problem in cross-modal retrieval. This method achieves robustness against query shift by adjusting query predictions and designing a joint objective function, which is an interesting and potentially influential direction for research.
2.	The authors have conducted extensive experiments on multiple datasets, including COCO-C and Flickr-C, to verify the effectiveness of the proposed method. The experiments cover comparisons across different model types and sizes, as well as varying severity levels of query shift, demonstrating the robustness of the method.
3.	The paper not only introduces a new method but also provides an in-depth analysis of the impact of query shift on cross-modal retrieval, revealing how query shift can reduce the uniformity of the query modality and increase the gap between the query and gallery modalities. These theoretical analyses offer valuable insights for future research.

**Weaknesses:**

1.	The TCR method proposed in the paper performs model adaptation at test time, which may increase additional computational costs. It is recommended that the authors analyze the computational complexity of the model and the additional cost incurred.
2.	Are the COCO-C and Flickr-C datasets constructed by the authors themselves? It seems that the paper does not explain whether the results of the baseline methods for comparison were obtained by the authors' own experiments or cited from their respective articles. If they were obtained through their own experiments, it should be clarified whether such comparisons are fair (whether they were trained on the new baselines), which is quite confusing for readers.
3.	The authors are advised to provide more explanation on the baselines and to include a few actual examples of query shifts to help readers intuitively feel the task.

**Questions:**

Please address my concerns proposed in Weakness.

---

> ### Author Response · Authors · 2024-11-21
>
> Thanks for the detailed comments. In the following, we will answer your questions one by one.
>
> >Q1: The TCR method proposed in the paper performs model adaptation at test time, which may increase additional computational costs. It is recommended that the authors **analyze the computational complexity of the model and the additional cost incurred**.
>
> **A1**: Thanks for your comments. In response to your insightful suggestion, we conduct additional experiments to analyze the efficiency of TCR. To this end, we choose the pre-trained model BLIP as the source model and perform zero-shot retrieval on the COCO dataset. We measure the GPU time during the test-time adaptation phase. Results are summarized in Table 19 within the revised manuscript. For your convenience, we attach the corresponding results in the following table.
>
> | Method | TR            | IR            | Avg.          |
> | ------ | ------------- | ------------- | ------------- |
> | Tent   | 285.5 seconds | 189.7 seconds | 237.6 seconds |
> | EATA   | 276.3 seconds | 190.4 seconds | 233.3 seconds |
> | DeYO   | 391.6 seconds | 254.2 seconds | 322.9 seconds |
> | Ours   | 291.1 seconds | 193.6 seconds | 242.4 seconds |
>
> Note that the learnable parameters of all the methods are the same for a fair comparison. The results underscore that TCR achieves adaptation more efficiently than the augmentation-based method DeYO[A]. Compared to the vanilla Tent[B] and EATA[C] (only low-entropy samples are employed for optimization), TCR requires only a negligible additional time cost, primarily due to the nearest neighbor selection in the query prediction refinement module.
>
> >Q2: Are the COCO-C and Flickr-C datasets constructed by the authors themselves? It seems that the paper does not explain whether the results of the baseline methods for comparison were obtained by the authors' own experiments or cited from their respective articles. If they were obtained through their own experiments, **it should be clarified whether such comparisons are fair (whether they were trained on the new baselines), which is quite confusing for readers**.
>
> **A2**: We apologize for the confusion arising from the initial presentation. Following the setting in [D], we have introduced 31 types of corruptions to establish the Query Shift benchmarks (i.e., COCO-C and Flickr-C datasets). The details of the benchmarks are provided in Appendix B.1 and the data processing code is available at **https://github.com/Jielin-Qiu/MM_Robustness**. Specifically, for image corruptions, we employ `image_perturbation/perturb_COCO_IP.py` to construct image corruptions for the COCO dataset and `image_perturbation/perturb_Flickr30K_IP.py` for the Flickr dataset. For text corruptions, we utilize `text_perturbation/perturb_COCO_TP.py` to construct text corruptions for the COCO dataset and `text_perturbation/perturb_Flickr30K_TP.py` for the Flickr dataset.
>
> In the paper, we aim to achieve online adaptation for cross-modal retrieval models under query shift, which is a novel challenge. Unfortunately, existing test-time adaptation (TTA) methods overlook the query shift in cross-modal settings and do not address this challenge. Moreover, **most existing TTA methods are specifically designed for the recognition task and cannot be directly employed for the cross-modal retrieval task**. To solve the problem, we **propose a simple baseline** so that the recognition-oriented TTA methods could be employed for the cross-modal retrieval task. **For a fair comparison, the optimizer, learning rate, and training parameters across all the methods are the same**. Besides, we carefully set the temperature for baselines on various datasets, as detailed in Appendix B.2. For your convenience, we attach the corresponding content as follows.
>
> To guarantee the performance of the baselines, **we select the optimal temperature (Eq. 1) for the TTA baselines upon each dataset**. According to Fig. 4(a), the temperature is fixed as $0.01$ for COCO-C, Flickr-C, COCO, Flickr, and Nocaps datatsets, $0.001$ for Fashion-Gen dataset, and $0.0001$ for CUHK-PEDE and ICFG-PEDES datasets.
>
> The code will be released upon acceptance of the paper.

---

> > ### Author Response · Authors · 2024-11-21
> >
> > >Q3: The authors are advised to provide **more explanation on the baselines** and to **include a few actual examples of query shifts** to help readers intuitively feel the task.
> >
> > **A3**: Thanks for your valuable suggestions. In Appendix B.1, **we have provided examples of query shift on the COCO-C dataset, which includes 16 types of image corruption and 15 types of text corruption.**
> >
> > **More examples of query shifts.** In response to your insightful suggestion, we offer more examples of query shifts in the ReID domain. **The examples are depicted in Fig. 6 within the revised manuscript**. For your convenience, we summarize the difference about the distribution shifts between the CUHK-PEDES dataset and ICFG-PEDES datasets:
> >
> > CUHK-PEDES is a dataset designed for text-to-image person re-identification, and the test set consists of 3,074 images and 6,156 textual descriptions associated with 1,000 identities. The images are sourced from five re-identification datasets, CUHK03 [E], Market-1501 [F], SSM [G], VIPER [H], and CUHK01 [I]. These images mainly capture outdoor scenes in diverse public spaces, such as markets and campuses. The textual descriptions often contain details not directly relevant to identity (e.g., actions and backgrounds), with an average of 23.5 words per description.
> >
> > ICFG-PEDES is a large-scale text-to-image person re-identification dataset, and the test set contains  19,848 image-text pairs of 1,000 identities. The images are sourced from the MSMT17 dataset [J] and depict scenes within a campus environment, with a mix of indoor and outdoor settings. Textual descriptions are more identity-focused and fine-grained, averaging 37.2 words per description.
> >
> > Notably, images in the ICFG-PEDES dataset are collected over multiple days at different times (morning, noon, and afternoon), which introduces considerable illumination variation. In contrast, images in the CUHK-PEDES dataset are sourced from short-duration surveillance videos, leading to minimal lighting variation.

---

> > > ### Author Response · Authors · 2024-11-21
> > >
> > > **More explanation on the baselines.** In response to your constructive feedback, we provide more detailed introdution of the baselines, , which can be found in Appendix B.3 due to space limitations. For your convenience, we attach the added statement as follows.
> > >
> > > Test-time Adaptation (TTA) aims to reconcile the distribution shifts in an online manner.  Towards achieving this goal, Fully TTA ([B]) has been proposed, which fine-tunes the BatchNorm layers by minimizing entropy during the test phase. EATA ([C]) employs a Fisher regularizer to limit excessive model parameter changes and filter out high-entropy samples via selection strategy. SAR [K] removes high-gradient samples and promotes flat minimum weights, enhancing robustness against more challenging TTA scenarios such as mixed domain shifts, single-sample adaptation, and imbalanced label shifts. READ ([L]) proposes a noise-robust adaptation loss and reliable fusion module to tackle the reliability bias challenge in the multi-modal setting. DeYO ([A]) reveals the unreliability of treating entropy as the confidence metric and establishes a novel metric by measuring the difference between predictions before and after applying an object-destructive transformation.
> > >
> > > **Reference:**
> > >
> > > [A] Jonghyun Lee, Dahuin Jung, Saehyung Lee, Junsung Park, Juhyeon Shin, Uiwon Hwang, and Sungroh Yoon. Entropy is not enough for test-time adaptation. In ICLR, 2024.
> > >
> > > [B] Dequan Wang, Evan Shelhamer, Shaoteng Liu, Bruno Olshausen, and Trevor Darrell. Tent: Fully test-time adaptation by entropy minimization. In ICLR, 2021.
> > >
> > > [C] Shuaicheng Niu, Jiaxiang Wu, Yifan Zhang, Yaofo Chen, Shijian Zheng, Peilin Zhao, and Mingkui Tan. Efﬁcient test-time model adaptation without forgetting. In ICML, 2022.
> > >
> > > [D] Jielin Qiu, Yi Zhu, Xingjian Shi, Florian Wenzel, Zhiqiang Tang, Ding Zhao, Bo Li, and Mu Li. *Benchmarking Robustness of Multimodal Image-Text Models under Distribution Shift*. Journal of Data-centric Machine Learning Research, 2023.
> > >
> > > [E] Wei Li, Rui Zhao, Tong Xiao, and Xiaogang Wang. DeepReID: Deep Filter Pairing Neural Network for Person Re-Identification. In CVPR, 2014.
> > >
> > > [F] Liang Zheng, Liyue Shen, Lu Tian, Shengjin Wang, Jiahao Bu, and Qi Tian. Person Re-identification Meets Image Search. In arXiv, 2015.
> > >
> > > [G] Tong Xiao, Shuang Li, Bochao Wang, Liang Lin, and Xiaogang Wang. Joint Detection and Identification Feature Learning for Person Search. In arXiv, 2016.
> > >
> > > [H] Douglas Gray, Shane Brennan, and Hai Tao. Evaluating appearance models for recognition, reacquisition, and tracking. In Proc. IEEE International Workshop on Performance Evaluation for Tracking and Surveillance (PETS), 2007.
> > >
> > > [I] Wei Li, Rui Zhao, and Xiaogang Wang. Human Reidentification with Transferred Metric Learning. In ACCV, 2012.
> > >
> > > [J] Longhui Wei, Shiliang Zhang, Wen Gao, and Qi Tian. Person Transfer GAN to Bridge Domain Gap for Person Re-Identification. In CVPR, 2018.
> > >
> > > [K] Shuaicheng Niu, Jiaxiang Wu, Yifan Zhang, Zhiquan Wen, Yaofo Chen, Peilin Zhao, and Mingkui Tan. Towards stable test-time adaptation in dynamic wild world. In ICLR, 2023.
> > >
> > > [L] Mouxing Yang, Yunfan Li, Changqing Zhang, Peng Hu, and Xi Peng. Test-time Adaption against Multi-modal Reliability Bias. In ICLR, 2024.

---

> > > > ### Comment · Reviewer_jDwD · 2024-11-24
> > > >
> > > > Thanks for the response, I choose to keep my score.

---

> > > > > ### Author Response · Authors · 2024-11-24
> > > > >
> > > > > We greatly appreciate the time and effort you invested in reviewing our work. Your feedback has been helpful in improving the paper. We would be happy to discuss further if needed.

---

### Official Review · Reviewer_t1ei · 2024-10-29

**Soundness:** 3
**Presentation:** 3
**Contribution:** 3
**Rating:** 8
**Confidence:** 3

**Summary:**

This paper introduces a novel setting, cross-modal retrieval under query shift. To address this challenge, it introduces a test-time adaptation method called TCR, which includes a query prediction refinement module to produce retrieval-optimized predictions for incoming queries. Additionally, it employs a joint objective function for online adaptation, effectively handling the query shift and noise.

**Strengths:**

1. The research question, cross-modal retrieval under query shift, is challenging and holds significant practical relevance.
2. Although this method builds on the principles of TTA, it also reveals TTA’s limitations in cross-modal retrieval and effectively overcomes these challenges.
3. Extensive experiments demonstrate the effectiveness of the proposed TCR method.
4. The paper is well-organized and well-written, enhancing the clarity and impact of its findings.

**Weaknesses:**

1. This research setting is limited by the assumption that each query batch contains i.i.d. samples. However, in real scenarios, query shift may occur unpredictably, introducing non-i.i.d. data within the same batch. This raises concerns about the method’s applicability under such conditions.
2. Regarding the emergence of query shift, I am curious whether temporal issues, such as temporal shifts or concept drift discussed in [1-3], are present in real-world scenarios. Could the authors provide relevant discussion on this aspect?
    [1] Evolving standardization for continual domain generalization over temporal drift. *NIPS 2023*.
    [2] Temporal domain generalization with drift-aware dynamic neural networks. *arXiv preprint arXiv:2205.10664* (2022)
    [3]Online Boosting Adaptive Learning under Concept Drift for Multistream Classification, AAAI 2024
3. In Section 3.2.1 on candidate selection, it would be valuable to address two points: first, whether gallery shift affects the outcomes of nearest neighbor selection; and second, how the number of selected candidates impacts the results. Additional experiments should be conducted to clarify these aspects.
4. In Section 3.2.2, given the shift between the source and target domains, it is unclear why source-domain-like data can be directly selected based on centers. Could the authors provide further analysis and explanation on this approach?
5. In section 3.5the definition of S(x_{i}^Q) in Equation (11) lacks corresponding theoretical analysis.
6. In the experiments, the authors employed various methods to generate image or text query shifts. I believe the results may depend on the specific shift generation techniques used. Therefore, it is crucial to provide access to the data processing methods and code to ensure the reproducibility of the experimental results.
7. I recommend that the authors discuss the limitations of the proposed method and outline specific future research directions. This would provide readers with additional insights and considerations for further exploration.

**Questions:**

discussed in Weaknesses section.

---

> ### Author Response · Authors · 2024-11-21
>
> Thanks for the insightful reviews. We will answer your questions one by one in the following.
>
> > **Q1**: This research setting is limited by the assumption that each query batch contains i.i.d. samples. However, in real scenarios, **query shift may occur unpredictably, introducing non-i.i.d. data within the same batch**. This raises concerns about the method’s applicability under such conditions.
>
> **A1**: In order to address your concerns, we conduct more experiments on the COCO-C benchmark, **investigating the robustness of the proposed TCR under non-i.i.d. settings** (i.e., **Mixed Severity Levels** and **Mixed Corruption Types**). Specifically,
>
> - Mixed Severity Levels: For each corruption, we create the test pairs by selecting $1/m$ of the data from each severity level, resulting in a total of $N$ test pairs, where $m$ is the number of severity levels and m=5 / 7 / 2  for the image / character-level / word-level corruptions.
>
> - Mixed Corruption Types: For the text retrieval, we construct the test pairs by selecting 1/16 of the data from each image corruption (1 through 16), resulting in a total of $N$ test pairs. For the image retrieval, we create test pairs by selecting 1/15 of the data from each text corruption (1 through 15), resulting in a total of $N$ test pairs.
>
> To verify the effectiveness of TCR under the Mixed Severity and Mixed Corruption Types settings, we choose the typical TTA method Tent ([A]) and the SOTA TTA methods EATA ([B]), DeYO ([C]) as baselines for comparisons. In the experiment, we carry out the experiments on the COCO-C benchmark, and the corresponding results are depicted in Tables 13-15 within the revised manuscript. For your convenience, we attach the corresponding numerical results (regarding Recall@1) in the following tables.
>
> | Mixed  Severity Levels | Gauss.   | Shot     | Impul.   | Speckle  | Defoc.   | Glass    | Motion   | Zoom     | Snow     | Frost    | Fog      | Brit.    | Contr.   | Elastic  | Pixel    | JPEG     | Avg.     |
> | ---------------------- | -------- | -------- | -------- | -------- | -------- | -------- | -------- | -------- | -------- | -------- | -------- | -------- | -------- | -------- | -------- | -------- | -------- |
> | BLIP ViT-B/16          | 61.8     | 61.8     | 59.7     | 66.4     | 58.5     | 70.7     | 56.7     | 22.5     | 42.6     | 60.4     | 66.0     | 70.8     | 61.0     | 61.2     | 47.8     | 69.8     | 58.6     |
> | Tent                   | 65.3     | 64.9     | 59.9     | 69.4     | 31.6     | 74.1     | 35.7     | 1.9      | 10.7     | 63.3     | 70.4     | 73.8     | 64.4     | 65.8     | 47.8     | 71.5     | 54.4     |
> | EATA                   | 64.9     | 65.6     | 64.6     | 70.0     | 62.0     | 74.3     | 61.7     | 28.1     | 55.3     | 63.9     | 71.1     | 74.4     | 65.5     | 66.0     | 53.7     | 72.7     | 63.4     |
> | DeYO                   | 64.0     | 66.0     | 63.0     | 69.8     | 64.6     | 74.6     | 63.0     | 5.8      | 56.1     | 65.7     | 71.4     | 74.5     | 65.7     | 67.8     | 52.5     | 72.7     | 62.3     |
> | Ours                   | **67.2** | **68.1** | **66.6** | **70.7** | **67.0** | **75.8** | **65.8** | **45.7** | **61.2** | **68.4** | **74.2** | **75.2** | **70.4** | **70.4** | **58.6** | **73.5** | **67.4** |
>
> | Mixed  Severity Levels | OCR      | CI       | CR       | CS       | CD       | SR       | RI       | RS       | RD       | IP       | Avg.     |
> | ---------------------- | -------- | -------- | -------- | -------- | -------- | -------- | -------- | -------- | -------- | -------- | -------- |
> | BLIP ViT-B/16          | 42.1     | 29.7     | 28.0     | 33.9     | 30.0     | 44.8     | 51.7     | 50.5     | 50.8     | 56.8     | 41.8     |
> | Tent                   | 42.4     | 28.5     | 23.6     | 33.8     | 26.9     | 45.5     | 52.4     | 51.4     | 50.9     | 57.0     | 41.2     |
> | EATA                   | 43.4     | 30.8     | 29.4     | 34.8     | 30.7     | 46.0     | 53.2     | 51.8     | 51.4     | 57.6     | 42.9     |
> | DeYO                   | 43.4     | 30.7     | 29.3     | 35.0     | 30.9     | 46.2     | 53.4     | 51.9     | 51.4     | 57.7     | 43.0     |
> | Ours                   | **44.4** | **32.2** | **30.6** | **35.7** | **31.7** | **46.3** | **53.8** | **52.1** | **51.5** | **57.4** | **43.6** |
>
> Note that for the Mixed Corruption Types setting, there are five levels of the mixed corruptions in text retrieval, corresponding to the image corruptions with five severity levels. For image retrieval, the severity levels of character-level / word-level / sentence-level text corruptions are 7 / 2 / 1. Thus, we select the two highest severity levels for character-level and word-level corruptions, and combine them with sentence-level corruptions, resulting in two levels of the mixed corruptions.

---

> > ### Author Response · Authors · 2024-11-21
> >
> > | Mixed Corruption Types | TR@1 Level 1 | TR@1 Level 2 | TR@1 Level 3 | TR@1 Level 4 | TR@1 Level 5 | IR@1 Level 1 | IR@1 Level 2 | Avg.     |
> > | ---------------------- | ------------ | ------------ | ------------ | ------------ | ------------ | ------------ | ------------ | -------- |
> > | BLIP ViT-B/16          | 68.8         | 64.5         | 61.4         | 54.5         | 44.9         | 42.5         | 41.1         | 53.9     |
> > | Tent                   | 70.0         | 67.0         | 64.4         | 56.4         | 33.2         | 42.2         | 38.5         | 53.1     |
> > | EATA                   | 71.7         | 68.2         | 64.7         | 58.9         | 48.0         | 43.2         | 41.9         | 56.7     |
> > | DeYO                   | 71.5         | 68.4         | 65.1         | 59.9         | 48.3         | 42.9         | 41.8         | 56.8     |
> > | Ours                   | **73.3**     | **70.4**     | **66.9**     | **61.5**     | **53.6**     | **43.8**     | **42.3**     | **58.8** |
> >
> > **The performance superiority of TCR over all baselines under both Mixed Severity and Mixed Corruption Types settings demonstrates its robustness against non-i.i.d query shift.**
> >
> > > Q2: Regarding the emergence of query shift, **I am curious whether temporal issues, such as temporal shifts or concept drift discussed in [1-3], are present in real-world scenarios**. Could the authors provide relevant discussion on this aspect?
> > > [1] Evolving standardization for continual domain generalization over temporal drift. *NIPS 2023*.
> > > [2] Temporal domain generalization with drift-aware dynamic neural networks. *arXiv preprint arXiv:2205.10664* (2022).
> > > [3] Online Boosting Adaptive Learning under Concept Drift for Multistream Classification. AAAI 2024.
> >
> > **A2**: Thanks for your constructive comments. We completely agree with your insightful opinion that the temporal issues would lead to query shift. **We have cited these related works and established some connections with them in Appendix E of the revised manuscript.** To be specific, as discussed in [D], the underlying distributions of data are distinct at different times, leading to **concept drift**. For instance, in weather forecasting, data is collected across diverse distributions (e.g., sunny, frost, snow). It is noteworthy that we have evaluated TCR under query shift caused by various weather conditions (e.g., frost, snow, fog). **The corresponding results from COCO-C (Table 1 in the manuscript) and Flickr-C (Table 7 in the manuscript) benchmarks demonstrate the robustness of TCR against concept drift to some extent**. For your convenience, we attach the corresponding numerical results (regarding TR@1) in the following tables.
> >
> > | COCO-C Benchmark       | Snow     | Frost     | Fog      | Bright     | Avg.     |
> > | ---------------------- | -------- | --------- | -------- | ---------- | -------- |
> > | BLIP ViT-B/16          | 32.3     | 52.2      | 57.0     | 66.8       | 52.1     |
> > | Tent                   | 31.9     | 48.7      | 56.3     | 66.5       | 50.9     |
> > | EATA                   | 45.6     | 56.7      | 62.5     | 71.4       | 59.0     |
> > | SAR                    | 38.0     | 56.2      | 59.1     | 70.6       | 56.0     |
> > | READ                   | 39.9     | 49.9      | 58.4     | 70.3       | 54.6     |
> > | DeYO                   | 37.5     | 59.7      | 66.4     | 71.2       | 58.7     |
> > | Ours                   | **56.5** | **64.1**  | **71.0** | **73.4**   | **66.3** |
> > | **Flickr-C Benchmark** | **Snow** | **Frost** | **Fog**  | **Bright** | **Avg.** |
> > | BLIP ViT-B/16          | 66.4     | 80.4      | 79.5     | 85.5       | 78.0     |
> > | Tent                   | 67.2     | 80.9      | 79.6     | 86.8       | 78.6     |
> > | EATA                   | 72.0     | 83.7      | 82.5     | 87.9       | 81.5     |
> > | SAR                    | 71.9     | 83.1      | 82.2     | 87.9       | 81.3     |
> > | READ                   | 71.7     | 83.8      | 81.9     | 87.7       | 81.3     |
> > | DeYO                   | 73.1     | 84.1      | 83.2     | 88.6       | 82.3     |
> > | Ours                   | **78.2** | **85.2**  | **85.7** | **89.5**   | **84.7** |

---

> > > ### Author Response · Authors · 2024-11-21
> > >
> > > As noted in [E] and [F], the collected data would continuously vary over time, resulting in **temporal shift**. For example, changes in lighting conditions throughout the day could impact the distribution of the collected data. For evaluation, we have conducted experiments under the CUHK2ICFG setting (Table 4 in the manuscript). Specifically, the ICFG-PEDES dataset is gathered at different times of the day (i.e., morning, noon, and afternoon), while the CUHK-PEDES dataset is derived from short-duration surveillance videos. Therefore, compared to CUHK-PEDES, the data in the ICFG-PEDES dataset exhibit distribution shifts due to time changes, such as illumination variation. The corresponding results from the CUHK2ICFG setting demonstrate that TCR could achieve robustness against temporal shift to some extent. For your convenience, we attach the corresponding numerical results (regarding IR@1) in the following tables.
> > >
> > > | Method        | CUHK2ICFG |
> > > | ------------- | --------- |
> > > | CLIP ViT-B/16 | 41.0      |
> > > | Tent          | 41.9      |
> > > | EATA          | 42.2      |
> > > | SAR           | 42.2      |
> > > | READ          | 42.3      |
> > > | DeYO          | 42.2      |
> > > | Ours          | **42.4**  |
> > >
> > > The three works mentioned are primarily designed for the classification task and focus on addressing distribution shifts caused by temporal issues. In contrast, TCR aims to tackle the query shift challenge in the cross-modal retrieval task. **Notably, any distribution shifts in the query modality would lead to query shift, not limited to temporal issues**. For instance, personalized issues such as writing habits and styles, as well as real-world corruptions like noise and blur, would lead to query shift.
> > > > Q3: In Section 3.2.1 on candidate selection, it would be valuable to address two points: first, **whether gallery shift affects the outcomes of nearest neighbor selection**; and second, **how the number of selected candidates impacts the results**. Additional experiments should be conducted to clarify these aspects.
> > >
> > > **A3**: Thanks for your comments. In the submission, we have conducted experiments under the Query-Gallery Shift setting (Table 3 in the manuscript), which demonstrates that TCR could improve retrieval performance even when the gallery modality occurs distribution shift. To address your concerns, we conduct more experiments during the rebuttal and present the results and analysis as follows.
> > >
> > > **Whether gallery shift affects the outcomes of nearest neighbor selection.** In response to your insightful suggestion, we conduct additional experiments to investigate whether gallery shift would affect the outcomes of nearest neighbor selection. Specifically, **we carry out experiments on the COCO-C benchmark under two gallery shift settings**, i.e., **only gallery shift** setting, **both query and gallery shift** setting. The corresponding results are depicted in Tables 16-17 within the revised manuscript. For your convenience, we attach the numerical results regarding Recall@1 and neighbor ACC (i.e., the cross-modal nearest neighbor of the query is correct) in the following tables. Notably, for the baseline BLIP ViT-B/16, the neighbor ACC and R@1 are the same since both are computed using cosine similarity for ranking.
> > >
> > > - Only Gallery Shift: In this setting, there is no distribution shift in the query modality. For the baseline BLIP ViT-B/16, the IR@1 and TR@1 without any query or gallery shift are 57.1% and 74.0%, respectively.
> > >
> > > | IR/Gallery Shift Types    | Gauss. | Shot | Impul. | Speckle | Defoc. | Glass | Motion | Zoom | Snow | Frost | Fog  | Brit. | Contr. | Elastic | Pixel | JPEG | Avg. |
> > > | ------------------------- | ------ | ---- | ------ | ------- | ------ | ----- | ------ | ---- | ---- | ----- | ---- | ----- | ------ | ------- | ----- | ---- | ---- |
> > > | BLIP ViT-B/16 (ACC & R@1) | 35.4   | 37.4 | 35.9   | 44.4    | 38.5   | 53.2  | 35.8   | 15.2 | 36.5 | 43.0  | 47.7 | 51.7  | 32.5   | 36.2    | 20.3  | 48.9 | 38.3 |
> > > | TCR (ACC)                 | 36.3   | 38.0 | 36.6   | 45.2    | 39.1   | 53.7  | 37.4   | 16.4 | 38.4 | 44.2  | 49.2 | 52.5  | 33.1   | 38.5    | 21.8  | 49.5 | 39.4 |
> > > | TCR (R@1)                 | 36.5   | 38.6 | 37.1   | 45.2    | 39.9   | 53.8  | 37.4   | 16.8 | 38.3 | 44.5  | 49.3 | 52.4  | 33.5   | 38.5    | 21.8  | 49.6 | 39.6 |

---

> > > > ### Author Response · Authors · 2024-11-21
> > > >
> > > > | TR/Gallery Shift Types    | OCR  | CI   | CR   | CS   | CD   | SR   | RI   | RS   | RD   | IP   | Formal | Casual | Passive | Active | Backtrans | Avg. |
> > > > | ------------------------- | ---- | ---- | ---- | ---- | ---- | ---- | ---- | ---- | ---- | ---- | ------ | ------ | ------- | ------ | --------- | ---- |
> > > > | BLIP ViT-B/16 (ACC & R@1) | 49.7 | 23.1 | 20.1 | 34.5 | 22.8 | 59.7 | 64.8 | 65.2 | 66.9 | 73.1 | 73.2   | 72.5   | 71.5    | 73.6   | 71.1      | 56.1 |
> > > > | TCR (ACC)                 | 55.5 | 26.8 | 23.5 | 39.5 | 27.5 | 66.2 | 70.9 | 70.4 | 71.3 | 77.1 | 76.5   | 76.2   | 75.1    | 77.3   | 75.4      | 60.6 |
> > > > | TCR (R@1)                 | 55.5 | 26.8 | 23.5 | 39.0 | 27.7 | 65.9 | 70.9 | 70.6 | 71.6 | 77.2 | 76.7   | 76.2   | 75.2    | 77.3   | 75.3      | 60.6 |
> > > >
> > > > - Both Query and Gallery Shift: In this setting, we choose the OCR / Gaussian corruptions as the query shift for image / text retrieval, respectively. For the baseline BLIP ViT-B/16, the IR@1 / TR@1 with OCR / Gaussian corruptions is 31.4% / 43.4%.
> > > >
> > > > | IR/Gallery  Shift Types   | Gauss. | Shot | Impul. | Speckle | Defoc. | Glass | Motion | Zoom | Snow | Frost | Fog  | Brit. | Contr. | Elastic | Pixel | JPEG | Avg. |
> > > > | ------------------------- | ------ | ---- | ------ | ------- | ------ | ----- | ------ | ---- | ---- | ----- | ---- | ----- | ------ | ------- | ----- | ---- | ---- |
> > > > | BLIP ViT-B/16 (ACC & R@1) | 18.5   | 19.8 | 18.6   | 23.4    | 20.1   | 28.8  | 19.0   | 8.2  | 18.9 | 22.5  | 25.5 | 27.6  | 17.1   | 18.6    | 10.4  | 26.0 | 20.2 |
> > > > | TCR (ACC)                 | 20.6   | 21.8 | 20.6   | 26.0    | 22.6   | 31.6  | 21.5   | 9.3  | 21.6 | 25.5  | 28.8 | 30.5  | 18.9   | 21.8    | 12.1  | 28.4 | 22.6 |
> > > > | TCR (R@1)                 | 20.6   | 21.8 | 20.6   | 26.0    | 22.6   | 31.7  | 21.5   | 9.4  | 21.6 | 25.6  | 28.8 | 30.5  | 19.0   | 21.8    | 12.2  | 28.4 | 22.6 |
> > > >
> > > > | TR/Gallery  Shift Types   | OCR  | CI   | CR   | CS   | CD   | SR   | RI   | RS   | RD   | IP   | Formal | Casual | Passive | Active | Backtrans | Avg. |
> > > > | ------------------------- | ---- | ---- | ---- | ---- | ---- | ---- | ---- | ---- | ---- | ---- | ------ | ------ | ------- | ------ | --------- | ---- |
> > > > | BLIP ViT-B/16 (ACC & R@1) | 27.2 | 13.2 | 11.1 | 18.4 | 27.2 | 32.5 | 36.3 | 37.2 | 38.9 | 42.2 | 42.8   | 42.7   | 41.0    | 43.2   | 41.0      | 33.0 |
> > > > | TCR (ACC)                 | 35.7 | 16.0 | 15.2 | 24.7 | 35.7 | 42.1 | 47.1 | 46.0 | 48.2 | 53.0 | 53.8   | 53.0   | 51.9    | 53.1   | 51.7      | 41.8 |
> > > > | TCR (R@1)                 | 35.2 | 16.5 | 15.3 | 24.6 | 35.2 | 42.2 | 47.7 | 45.4 | 48.3 | 52.8 | 52.4   | 52.7   | 52.1    | 53.5   | 52.1      | 41.7 |
> > > >
> > > > From the results, one could observe that **gallery shift degrades both retrieval performance and nearest neighbor selection accuracy**, whether in only gallery shift or both query and gallery shift settings. However, **the proposed TCR improves the retrieval performance under gallery shift, with the selected nearest neighbors more likely to be correct**. Besides, even under gallery shift setting, TCR could enhance retrieval performance surpassing the baseline performance without gallery shift. For example, in the both query and gallery shift setting, the text retrieval performance of TCR under RI (47.7%), RS (45.4%), Formal (52.4%), and Passive (52.1%) gallery shift exceeds the baseline performance without gallery shift (43.4%). It’s worth noting that in real-world scenarios, data with only gallery shift is rare, as the data in the gallery is often extensive and curated. In contrast, the queries of the users are more diverse, which might lead to the distribution shift challenge.

---

> ### Author Response · Authors · 2024-11-21
>
> **How the number of selected candidates impacts the results.** We conduct more experiments to investigate how the number of selected candidates affects performance. To this end, we directly perform zero-shot retrieval experiment on the COCO dataset with pre-trained BLIP as the source model. In the paper, we retrieve the most similar sample from the gallery set for each query, thus the number of selected candidates is equal to the batch size $B$. In the additional experiment, we vary the number of selected candidates at different values, i.e., $[0.2B, 0.5B, B, 2B, 5B, 10B, 50B]$. Specifically, assume that the number of selected candidates is $\lambda B$, where $\lambda$ is an integer. When $\lambda < 1$, we randomly select $\lambda B$ candidates from the original $B$ selected candidates. When $\lambda \geq 1$, we retrieve the most similar $\lambda$ candidates from the gallery set for each query, forming a new set of $\lambda B$ selected candidates. The results are depicted in Table 18 within the revised manuscript. For your convenience, we attach the corresponding numerical results in the following tables.
>
> | Number | 0.2B | 0.5B | B (Default) | 2B   | 5B   | 10B  | 50B  |
> | -------------------------------------- | ---- | ---- | ----------- | ---- | ---- | ---- | ---- |
> | TR@1                                   | 67.2 | 68.5 | 68.9        | 65.3 | 64.9 | 64.7 | 64.6 |
> | IR@1                                   | 47.3 | 48.3 | 48.9        | 48.3 | 48.2 | 48.0 | 47.5 |
>
> From the results, one could observe that **enlarging the number of selected candidates would significantly degrade the performance**. Such a phenomenon indicates that an excessively large number of candidates may lead to the underfitting issue, which **highlights the necessity and effectiveness of the query refinement module**.
>
> >Q4: In Section 3.2.2, given the shift between the source and target domains, **it is unclear why source-domain-like data can be directly selected based on centers**. Could the authors provide further analysis and explanation on this approach?
>
> **A4**: We appreciate your feedback. As illustrated in Fig. 1(c), we observe that distribution shift would diminish the modality uniformity, defined as the average distance between all samples and the modality center (Eq. 14 in the manuscript). For your convenience, we have attached the corresponding equation below.
> $$
>     \text{Uniformity}=\frac{1}{N^{Q}}\sum_{i=1}^{N^{Q}}\|\mathbf{z}_{i}^{Q}-\overline{\mathbf{Z}}^{Q}\|.
> $$
> In other words, the distribution shift would narrow the distance between samples and their modality centers. Therefore, we conclude that data from the source domain should exhibit higher modality uniformity. Based on the conclusion, we select samples farther from their modality centers as the source-domain-like data, since these samples enjoy higher modality uniformity.
> >Q5: In section 3.5，the definition of $S(x_{i}^Q)$ in Equation (11) **lacks corresponding theoretical analysis**.
>
> **A5**: We apologize for the initial oversight regarding the theoretical analysis in Section 3.5. The proposed noise-robust adaptation loss (Eq. 11) aims to **achieve robustness against heavy noise by excluding high-entropy query predictions from adaptation and assigning higher weights to query predictions with lower uncertainties**. Specifically, for a given query sample $x_{i}^{Q}$, let its entropy be denoted as $E(x_{i}^Q)$.
>
> When $E(x_i^Q) \geq E_m$, the weight $S(x_i^Q)$ is defined as
> $$
> S(x_i^Q) = 0.
> $$
> In this case, $x_{i}^{Q}$ is excluded from optimization. Such a design prevents high-entropy (i.e., noisy) query predictions from degrading performance, as their gradients produced by entropy loss might be biased and unreliable.
>
> When $0 \leq E(x_i^Q) < E_m$, the weight $S(x_i^Q)$ is positive and defined as
> $$
> S(x_i^Q) = 1 - \frac{E(x_i^Q)}{E_m}.
> $$
>
> The weight is inversely proportional to entropy, i.e., the weight decreases as entropy increases. Formally,
> $$
> S(x_i^Q) \propto \frac{1}{E(x_i^Q)}.
> $$
> The adaptive weighting strategy enjoys two advantages. On the one hand, query predictions with lower entropy (i.e., reliable predictions) are assigned with higher weights, thus guiding the optimization. On the other hand, query predictions with higher entropy (i.e., uncertain predictions) are assigned with lower weights, thereby preventing overfitting on noisy query predictions.

---

> > ### Author Response · Authors · 2024-11-21
> >
> > >Q6: It is crucial to **provide access to the data processing methods and code** to ensure the reproducibility of the experimental results.
> >
> > **A6**: Following the setting in [G], we have introduced 31 types of corruption to establish the Query Shift benchmarks (i.e., COCO-C and Flickr-C datasets). **The details of the benchmarks are provided in Appendix B.1** and the data processing code is available at **https://github.com/Jielin-Qiu/MM_Robustness**. Specifically, for image corruptions, we employ `image_perturbation/perturb_COCO_IP.py` to construct image corruptions for the COCO dataset and `image_perturbation/perturb_Flickr30K_IP.py` for the Flickr dataset. For text corruptions, we utilize `text_perturbation/perturb_COCO_TP.py` to construct text corruptions for the COCO dataset and `text_perturbation/perturb_Flickr30K_TP.py` for the Flickr dataset.
> >
> > >Q7: I recommend that the authors discuss **the limitations of the proposed method and outline specific future research directions**. This would provide readers with additional insights and considerations for further exploration.
> >
> > **A7**: Thanks for your valuable suggestions. In response to your concern, we summarize the following limitations and potential directions for future work of TCR.
> >
> > **Limitations.** The proposed TCR might have the following two limitations. On the one hand, although TCR achieves significant performance improvement in the cross-modal retrieval task, it remains uncertain whether TCR could achieve similar success in other cross-modal tasks, such as image captioning and visual question answering (VQA). On the other hand, the robustness of TCR against more challenging TTA scenarios (e.g., single-sample adaptation and continuous adaptation) is worth further investigation.
> >
> > **Future research**. In the future, we plan to extend TCR to more applications and more challenging scenarios. Specifically, we would like to extend TCR for a broader range of cross-modal tasks, such as image captioning and visual question answering (VQA). Besides, although we have demonstrated that TCR could achieve robustness against the query shift evaluated in the paper, further work is needed to verify whether TCR could address more challenging scenarios, such as temporal shift and concept drift.
> >
> > **Reference:**
> >
> > [A] Dequan Wang, Evan Shelhamer, Shaoteng Liu, Bruno Olshausen, and Trevor Darrell. Tent: Fully test-time adaptation by entropy minimization. In ICLR, 2021.
> >
> > [B] Shuaicheng Niu, Jiaxiang Wu, Yifan Zhang, Yaofo Chen, Shijian Zheng, Peilin Zhao, and Mingkui Tan. Efﬁcient test-time model adaptation without forgetting. In ICML, 2022.
> >
> > [C] Jonghyun Lee, Dahuin Jung, Saehyung Lee, Junsung Park, Juhyeon Shin, Uiwon Hwang, and Sungroh Yoon. Entropy is not enough for test-time adaptation. In ICLR, 2024.
> >
> > [D] En Yu, Jie Lu, Bin Zhang, and Guangquan Zhang. Online Boosting Adaptive Learning under Concept Drift for Multistream Classification. In AAAI, 2024.
> >
> > [E] Guangji Bai, Chen Ling, and Liang Zhao. Temporal domain generalization with drift-aware dynamic neural networks. In arXiv, 2022.
> >
> > [F] Mixue Xie, Shuang Li, Longhui Yuan, Chi Harold Liu, and Zehui Dai. Evolving standardization for continual domain generalization over temporal drift. In NIPS, 2023.
> >
> > [G] Jielin Qiu, Yi Zhu, Xingjian Shi, Florian Wenzel, Zhiqiang Tang, Ding Zhao, Bo Li, and Mu Li. *Benchmarking Robustness of Multimodal Image-Text Models under Distribution Shift*. Journal of Data-centric Machine Learning Research, 2023.

---

> > > ### Comment · Reviewer_t1ei · 2024-11-21
> > >
> > > Thank you very much for your responses. I have carefully reviewed each of them. You have addressed all my concerns, and I think this is a very interesting and meaningful work that provides valuable insights into cross-modal retrieval research. Therefore, I will increase my score.

---

> > > > ### Author Response · Authors · 2024-11-21
> > > >
> > > > Thank you for your prompt response and for upgrading your score! We deeply appreciate the time and effort you dedicated to reviewing our work. Your constructive feedback has been invaluable in helping us refine and improve the paper.

---

### Official Review · Reviewer_E6xm · 2024-11-03

**Soundness:** 3
**Presentation:** 4
**Contribution:** 3
**Rating:** 8
**Confidence:** 3

**Summary:**

The paper presents a Test-time adaptation for Cross-modal Retrieval (TCR) method to address query shift, which is a critical and understudied problem in cross-modal retrieval tasks. Query shift occurs when the distribution of online query streams differs from the source domain, leading to performance degradation in existing models. TCR introduces a query prediction refinement module and a joint objective function to refine query predictions and prevent query shift from disturbing the common space. It improves the existing test-time adaptation (TTA) methods with the capacity to manipulate both the modality uniformity and modality gap. Overall speaking, this paper is well-organized and of practical value.

**Strengths:**

The proposed TCR method addresses an important problem and is supported by strong experimental results.

It provides extensive experiments demonstrating the effectiveness of TCR against query shift. The comparisons with existing TTA methods show convincing improvements, with is a strong validation of the ablation study .

**Weaknesses:**

In Section 4.2, it is said that “We compare TCR with five SOTA TTA methods (Tent (Wang et al., 2021),  EATA(Niu et al.,2022), SAR(Niu et al.,2023), READ(Yang et al.,2024), and DeYO...”. These methods should be introduced in Section 2.2 of the related work part.

Line 212, ,where Q and G denotes as query modality and gallery modality for clarity in the following. Change to “denote””

Tables 1 and 2 appear too early. They should not be on Page 7 but on the page where they are referred for the first time,

**Questions:**

see above comment

---

> ### Author Response · Authors · 2024-11-21
>
> Thanks for your valuable reviews. We would like to address your concerns one by one in the following.
>
> > Q1: In Section 4.2, it is said that “We compare TCR with five SOTA TTA methods (Tent (Wang et al., 2021), EATA(Niu et al.,2022), SAR(Niu et al.,2023), READ(Yang et al.,2024), and DeYO...”. **These methods should be introduced in Section 2.2 of the related work part**.
>
> **A1**: Thanks for your valuable suggestions. We apologize for the missing details on the baselines. In response to your constructive feedback, we provide more detailed introdution of the baselines, which can be found in Appendix B.3 due to space limitations. For your convenience, we attach the added statement as follows.
>
> Test-time Adaptation (TTA) aims to reconcile the distribution shifts in an online manner.  Towards achieving this goal, fully TTA ([A]) has been proposed, which fine-tunes the BatchNorm layers by minimizing entropy during the test phase. EATA ([B]) employs a Fisher regularizer to limit excessive model parameter changes and filter out high-entropy samples via the selection strategy. SAR [C] removes high-gradient samples and promotes flat minimum weights, enhancing robustness against more challenging TTA scenarios such as mixed domain shifts, single-sample adaptation, and imbalanced label shifts. READ ([D]) proposes a noise-robust adaptation loss and reliable fusion module to tackle the reliability bias challenge in the multi-modal setting. DeYO ([E]) reveals the unreliability of treating entropy as the confidence metric and establishes a novel metric by measuring the difference between predictions before and after applying an object-destructive transformation.
>
> >Q2: Line 212, ,where Q and G denotes as query modality and gallery modality for clarity in the following. Change to “denote”
>
> **A2**: Thanks for your careful reading. We apologize for the typos and have revised them in the updated manuscript.
>
> >Q3: Tables 1 and 2 appear too early. They should not be on Page 7 but on the page where they are referred for the first time.
>
> **A3**: Thanks for your valuable suggestions. We apologize for the misplacement of Table 1 and Table 2 in the submission and have revised them in the updated manuscript.
>
> **Reference:**
>
> [A] Dequan Wang, Evan Shelhamer, Shaoteng Liu, Bruno Olshausen, and Trevor Darrell. Tent: Fully test-time adaptation by entropy minimization. In ICLR, 2021.
>
> [B] Shuaicheng Niu, Jiaxiang Wu, Yifan Zhang, Zhiquan Wen, Yaofo Chen, Peilin Zhao, and Mingkui Tan. Towards stable test-time adaptation in dynamic wild world. In ICLR, 2023.
>
> [C] Shuaicheng Niu, Jiaxiang Wu, Yifan Zhang, Yaofo Chen, Shijian Zheng, Peilin Zhao, and Mingkui Tan. Efﬁcient test-time model adaptation without forgetting. In ICML, 2022.
>
> [D] Mouxing Yang, Yunfan Li, Changqing Zhang, Peng Hu, and Xi Peng. Test-time Adaption against Multi-modal Reliability Bias. In ICLR, 2024.
>
> [E] Jonghyun Lee, Dahuin Jung, Saehyung Lee, Junsung Park, Juhyeon Shin, Uiwon Hwang, and Sungroh Yoon. Entropy is not enough for test-time adaptation. In ICLR, 2024.

---

> > ### Comment · Reviewer_E6xm · 2024-11-26
> >
> > Thanks for the authors's feedback.
> >
> > Compared methods in the experiments should be most relevant to the proposed work, but even their names did not appear in the related work part in the main text. I can understand due to page limit their details can be moved to appendix, but they should be at least mentioned in related work.

---

> > > ### Author Response · Authors · 2024-11-27
> > >
> > > Thanks for your valuable suggestions. The five baselines compared in the manuscript (Tent, EATA, SAR, READ, and DeYO) are the most relevant works to the proposed TCR. Specifically, Tent is the most classic TTA method; EATA and SAR are robust TTA methods designed to handle noisy predictions; DeYO is the state-of-the-art method for unimodal recognition, while READ is the state-of-the-art for multimodal recognition. Since the proposed TCR might be one of the first TTA works for cross-modal retrieval, we select the above baselines for comparison.
> > >
> > > In response to your insightful suggestion, we have cited these methods in Related Work (Sec. 2.2 in the manuscript). For your convenience, we attach the revised content as follows.
> > >
> > > To avoid the reduplicated training cost of the source model, Fully Test-Time Adaptation paradigm (Tent) has been proposed, which could be coarsely divided into the following three categories: i) online TTA methods (e.g., DeYO), which continually update the normalization layers by resorting to the unsupervised objectives, such as entropy minimization or its variants. ii) robust TTA methods (e.g., EATA, SAR), which strive to improve the robustness against noisy predictions, mixed distribution shifts, label shifts, and so on. iii) TTA beyond recognition, which focuses on the tasks including but not limited to image restoration, multimodal recognition (e.g., READ), and multimodal segmentation.
> > >
> > > We would be happy to discuss further if needed.

---

### Official Review · Reviewer_Wks8 · 2024-11-05

**Soundness:** 3
**Presentation:** 3
**Contribution:** 3
**Rating:** 6
**Confidence:** 3

**Summary:**

The paper addresses the challenge of cross-modal retrieval in scenarios where the query data distribution deviates from the source domain, a phenomenon known as "query shift." This deviation often leads to a performance decline in cross-modal retrieval systems. The authors propose a novel approach called TCR: Test-time adaptation for Cross-modal Retrieval, which adapts cross-modal retrieval models during inference to account for query shift. The proposed method includes a query prediction refinement module and a joint objective function to prevent the disturbances caused by the query shift, enhancing the uniformity within the query modality and minimizing the gap between query and gallery modalities. The model is designed to operate effectively in real time by adapting to changing online queries. The approach was tested on six popular image-text datasets and demonstrated superior performance against existing test-time adaptation (TTA) techniques.

**Strengths:**

1) The paper tackles the underexplored problem of query shift in cross-modal retrieval, providing a comprehensive analysis of its effects on retrieval performance. The method's unique combination of query prediction refinement and multiple loss functions sets it apart from traditional TTA approaches.
2) The authors conducted extensive experiments across six datasets and compared their method against several state-of-the-art TTA models. The amount of experiments is fair and convincing.
3) The paper proposes a joint objective consisting of three loss functions—uniformity learning, gap minimization, and noise-robust adaptation—that each address specific challenges introduced by query shift. This is a novel design for this problem.

**Weaknesses:**

1)  The authors provide only limited discussion regarding the sensitivity of the various hyperparameters involved, such as the temperature and trade-off parameters. A more detailed analysis would improve understanding of the model's adaptability to different scenarios.
2) The approach heavily relies on pre-trained models and assumes the existence of a well-aligned common space. In cases where the source domain model lacks robust representations, the effectiveness of TCR may be diminished. This could limit the generalizability of the approach to pre-trained models of different quality. More discussions and insights need to be given in order to make the paper more readable.

**Questions:**

1) The proposed TCR method aims to enhance retrieval robustness under query shift by manipulating modality uniformity and the modality gap. Given the variety of potential shifts in real-world data (e.g., subtle cultural variations, extreme distortions, rare domain-specific content), how does TCR perform across these different types of shifts?

2) Could the model's performance degrade if it encounters shifts it was not explicitly evaluated against? A thorough breakdown of the model’s robustness to a diverse set of query shifts would strengthen the understanding of its general applicability.

3) This paper introduces several hyperparameters, such as the temperature parameter (τ) for controlling the trade-off between smoothness and sharpness, and others for balancing the different loss terms. How sensitive is TCR to these hyperparameters, and how easy is it to tune them for new domains? Some more results from these ablations studies will be very beneficial.

---

> ### Author Response · Authors · 2024-11-21
>
> Thanks for your constructive reviews and suggestions. In the following, we will answer your questions one by one.
>
> > Q1: Given the variety of potential shifts in real-world data (e.g., subtle cultural variations, extreme distortions, rare domain-specific content), **how does TCR perform across these different types of shifts**?
>
> **A1**: Thanks for your constructive comments. In the submission, we have reported the experiment results on a variety of real-world query shift, including noise, blur, and weather (e.g., frost, snow, and fog) distortions. Besides, we have evaluated TCR on the rare domains, such as ReID domain, e-commerce domain and natural image domain. Notably, we have carried out the experiments to verify that TCR could handle the maximum severity level of the corruptions (Tables 1-2 and Tables 7-8). In other words, **TCR is able to address the most extreme distortions, which are highlighted and widely acknowledged in previous works** ([A] [B] [C]).
>
> In response to your insightful suggestion, **we conduct additional experiments in the even rarer remote sensing domain**. Specifically, we choose the BLIP as the source model and perform zero-shot retrieval on the remote sensing datasets RSICD ([D]) and RSITMD ([E]). To verify the effectiveness of TCR, we choose the typical TTA method Tent ([A]), the SOTA TTA methods EATA ([F]) and DeYO ([G]) as the baselines for comparisons. Results are summarized in Table 20 within the revised manuscript. For your convenience, we attach the corresponding results in the following tables.
>
> | Base2RSICD    | TR@1    | IR@1    |
> | ------------- | ------- | ------- |
> | BLIP ViT-B/16 | 6.4     | 6.8     |
> | Tent          | 5.7     | 5.4     |
> | EATA          | 6.9     | 6.7     |
> | DeYO          | 6.4     | 6.5     |
> | Ours          | **8.5** | **7.1** |
>
> | Base2RSITMD   | TR@1    | IR@1     |
> | ------------- | ------- | -------- |
> | BLIP ViT-B/16 | 7.6     | 10.4     |
> | Tent          | 7.9     | 9.3      |
> | EATA          | 8.0     | 10.4     |
> | DeYO          | 7.7     | 10.0     |
> | Ours          | **8.4** | **10.7** |
>
> **The results indicate that TCR could also achieve the best performance in even rarer remote sensing domain.**
>
> >Q.2: **Could the model's performance degrade if it encounters shifts it was not explicitly evaluated against**? A thorough breakdown of the model’s robustness to a diverse set of query shifts would strengthen the understanding of its general applicability.
>
> **A.2**: Thanks for your comments. In this paper, we have evaluated TCR on the Flickr and COCO datasets with image and text corruptions, which simulate real-world query shift. It is worth noting that we have introduced **a total of 130 perturbations across various severity levels**, comprising 80 for the image modality and 50 for the text modality. Specifically, the image modality includes 16 types of corruption, each with 5 levels of severity, while the text modality comprises 15 corruption types with 7/2/1 severity levels for character-level/word-level/sentence-level corruptions. Besides, we conduct experiments on the datasets with real-world query shift, including the ReID, e-commerce, and natural image domains.
>
> In Appendix D.3 of the manuscript, we have reported the experiment results on the 130 perturbations in Fig. 7 and Fig. 8. Specifically, we carry out the experiments on the COCO-C benchmark and report the average performance across various severity levels for each corruption. For your convenience, we have included a summary of these results (regarding Recall@1) in the following tables.

---

> > ### Author Response · Authors · 2024-11-21
> >
> > | BLIP  ViT-B/16 | Gauss.   | Shot     | Impul.   | Speckle  | Defoc.   | Glass    | Motion   | Zoom     | Snow     | Frost    | Fog      | Brit.    | Contr.   | Elastic  | Pixel    | JPEG     | Avg.     |
> > | -------------- | -------- | -------- | -------- | -------- | -------- | -------- | -------- | -------- | -------- | -------- | -------- | -------- | -------- | -------- | -------- | -------- | -------- |
> > | Tent           | 64.3     | 64.4     | 58.8     | 70.2     | 50.3     | 74.0     | 45.8     | 20.6     | 29.8     | 63.4     | 68.6     | 73.9     | 59.7     | 66.9     | 42.7     | 72.5     | 57.9     |
> > | EATA           | 64.1     | 65.5     | 62.1     | 70.3     | 64.2     | 74.9     | 62.6     | 26.9     | 55.5     | 64.4     | 70.8     | 74.6     | 66.4     | 67.5     | 52.5     | 72.8     | 63.4     |
> > | SAR            | 63.9     | 65.3     | 61.8     | 69.8     | 60.6     | 74.2     | 58.2     | 22.9     | 47.7     | 64.0     | 69.3     | 74.0     | 63.0     | 67.5     | 51.7     | 72.2     | 61.6     |
> > | READ           | 64.6     | 63.9     | 59.1     | 69.0     | 58.5     | 74.0     | 61.0     | 22.2     | 49.9     | 62.0     | 69.0     | 73.6     | 63.0     | 65.8     | 48.6     | 71.9     | 61.0     |
> > | DeYO           | 65.4     | 66.3     | 64.2     | 70.2     | 62.9     | 74.6     | 61.2     | 22.5     | 52.8     | 65.5     | 71.9     | 74.3     | 66.0     | 67.7     | 50.7     | 72.7     | 63.1     |
> > | Ours           | **67.6** | **67.8** | **67.1** | **71.2** | **67.8** | **75.8** | **66.2** | **46.4** | **62.1** | **68.6** | **74.0** | **75.6** | **71.1** | **71.8** | **59.3** | **73.2** | **67.8** |
> >
> > | BLIP  ViT-B/16 | OCR      | CI       | CR       | CS       | CD       | SR       | RI       | RS       | RD       | IP       | Formal   | Casual   | Passive  | Active   | Backtrans | Avg.     |
> > | -------------- | -------- | -------- | -------- | -------- | -------- | -------- | -------- | -------- | -------- | -------- | -------- | -------- | -------- | -------- | --------- | -------- |
> > | Tent           | 42.0     | 27.1     | 24.5     | 31.1     | 27.8     | 45.5     | 52.8     | 51.3     | 51.1     | 57.0     | 56.6     | 56.4     | 55.3     | 57.2     | 54.0      | 46.0     |
> > | EATA           | 43.3     | 30.5     | 29.0     | 34.6     | 30.7     | 46.2     | 53.2     | 51.6     | 51.3     | 56.6     | 56.6     | 56.3     | 55.6     | 57.2     | 54.2      | 47.1     |
> > | SAR            | 42.0     | 29.7     | 28.3     | 34.0     | 30.1     | 45.0     | 51.8     | 50.5     | 50.9     | 56.8     | 56.5     | 56.2     | 54.9     | 56.8     | 54.2      | 46.5     |
> > | READ           | 42.9     | 30.0     | 28.6     | 34.2     | 30.2     | 45.8     | 53.2     | 51.7     | 51.4     | 56.7     | 56.5     | 56.2     | 55.2     | 57.0     | 53.9      | 46.9     |
> > | DeYO           | 43.4     | 29.6     | 28.2     | 34.5     | 30.4     | 46.2     | 53.6     | 51.7     | 51.3     | 56.7     | 56.6     | 56.3     | 55.6     | 57.1     | 54.2      | 47.0     |
> > | Ours           | **44.0** | **31.7** | **30.3** | **35.2** | **31.5** | **46.6** | **54.0** | **52.0** | **51.6** | **57.3** | **57.1** | **56.8** | **56.0** | **57.3** | **54.7**  | **47.7** |
> >
> > **The results demonstrate that TCR outperforms all baselines across various severities and corruptions, showcasing its robustness against different distribution shifts.**
> >
> > Besides, we conduct more experiments on the remote sensing datasets RSICD ([D]) and RSITMD ([E]), which might encounter the query shift not explicitly evaluated in the paper. For your convenience, we have included a summary of these results in the following tables.
> >
> > | Base2RSICD    | TR@1    | IR@1    |
> > | ------------- | ------- | ------- |
> > | BLIP ViT-B/16 | 6.4     | 6.8     |
> > | Tent          | 5.7     | 5.4     |
> > | EATA          | 6.9     | 6.7     |
> > | DeYO          | 6.4     | 6.5     |
> > | Ours          | **8.5** | **7.1** |
> >
> > | Base2RSITMD   | TR@1    | IR@1     |
> > | ------------- | ------- | -------- |
> > | BLIP ViT-B/16 | 7.6     | 10.4     |
> > | Tent          | 7.9     | 9.3      |
> > | EATA          | 8.0     | 10.4     |
> > | DeYO          | 7.7     | 10.0     |
> > | Ours          | **8.4** | **10.7** |
> >
> > The results underscore that TCR achieves superior performance on the RSICD, RSITMD datasets from the remote sensing domain, further validating its effectiveness on unevaluated query shifts in the paper.

---

> ### Author Response · Authors · 2024-11-21
>
> >Q.3: This paper introduces several hyperparameters, such as the temperature parameter (τ) for controlling the trade-off between smoothness and sharpness, and others for balancing the different loss terms. **How sensitive is TCR to these hyperparameters, and how easy is it to tune them for new domains**? Some more results from these ablations studies will be very beneficial.
>
> **A.3**: Thanks for your comments. There are two hyperparameters in the paper, i.e., the temperature $\tau$ for controlling the trade-off between smoothness and sharpness, the trade-off parameter $t$  for controlling the intra-modality uniformity. For parameter sensitivity analysis, we have conducted ablation studies about $\tau$ and $t$ in Fig. 4(a) and Fig. 4(b), respectively. For your convenience, we attach the corresponding numerical results of Fig. 4(a) in the following tables.
>
> | $\tau$       | 1e-5 | 1e-4 | 1e-3 | 5e-3 | 0.01 | 0.02 | 0.05 | 0.1  |
> | ------------ | ---- | ---- | ---- | ---- | ---- | ---- | ---- | ---- |
> | Base2COCO    | 52.4 | 53.0 | 52.9 | 55.7 | 56.4 | 57.8 | 57.0 | 55.2 |
> | ICFG2CUHK    | 32.9 | 33.9 | 34.5 | 34.6 | 35.8 | 36.2 | 35.9 | 34.7 |
> | Flickr-C     | 58.8 | 62.3 | 64.0 | 66.0 | 67.3 | 69.1 | 65.2 | 64.7 |
> | Base2Fashion | 21.1 | 21.5 | 23.1 | 25.0 | 25.8 | 26.4 | 24.7 | 13.7 |
>
> **The results denote that TCR demonstrates stable performance within the range of [0.001,0.05] and achieves the best performance when $\tau=0.02$.**
>
> In response to your concern, **we conduct more ablations studies on the trade-off parameter $t$ under "Flickr-C", "ICFG2CUHK", "Base2Fashion" settings**. The results are summarized in Fig. 4(b) within the updated manuscript. For your convenience, we attach the corresponding numerical results (regarding Recall@1) in the following tables.
>
> | t            | 0.1  | 1.0  | 2.0  | 5.0  | 10.0 | 20.0 | 100.0 |
> | ------------ | ---- | ---- | ---- | ---- | ---- | ---- | ----- |
> | Base2COCO    | 58.4 | 58.4 | 58.8 | 58.9 | 59.0 | 58.5 | 58.0  |
> | ICFG2CUHK    | 37.0 | 37.1 | 37.3 | 37.3 | 37.3 | 37.3 | 37.0  |
> | Flickr-C     | 67.8 | 68.2 | 68.2 | 68.4 | 68.4 | 68.2 | 67.9  |
> | Base2Fashion | 26.9 | 27.0 | 27.0 | 27.1 | 27.4 | 27.1 | 27.0  |
>
> **The results indicate that TCR is not sensitive to the choice of the parameter $t$.**
>
> Besides, we apologize for the omission of the sensitivity analysis for these hyperparameters. In the revised manuscript, we have supplemented the detailed parameter analysis. For your convenience, we attach the added statement as follows.
>
> As shown in Fig. 4(a), we observe that: i) selecting an appropriate temperature for the existing TTA approach across various datasets is challenging; ii) even a low temperature (e.g., $1e-4$) is a better setting across all datasets, the performance degrades as a low temperature tends to make model overfitting on noisy query prediction. In contrast, the query prediction refinement module not only stabilizes the temperature setting for all the datasets but also prevents the model from either underfitting or overfitting by excluding some irrelevant samples in the gallery. Besides, TCR demonstrates stable performance within the range of [0.001,0.05] and achieves the best performance when $\tau=0.02$. As depicted in Fig. 4(b), one could observe that TCR is not sensitive to the choice of $t$.

---

> > ### Author Response · Authors · 2024-11-21
> >
> > >Q4: The approach heavily relies on pre-trained models and assumes the existence of a well-aligned common space. **In cases where the source domain model lacks robust representations, the effectiveness of TCR may be diminished**. This could limit the generalizability of the approach to pre-trained models of different quality. More discussions and insights need to be given in order to make the paper more readable.
> >
> > **A4**: Thanks for your comments. We acknowledge that TCR relies on the well-aligned common space, which is essential for achieving good intra-modality uniformity and inter-modality gap. In other words, if the source model is suboptimal, the performance improvements of TCR might be less pronounced. However, **we highlight that TCR outperforms all baselines and achieves a stable performance improvement across various pre-trained model types and sizes**, including BLIP ViT-B/16, BLIP ViT-L/16, CLIP ViT-B/16, and CLIP ViT-B/32. **Indeed, any test-time adaptation paradigm heavily depends on the high-quality source model** [A] [B] [F], i.e., the high-quality source model would provide more reliable predictions that support performance improvement.
> >
> > **Reference:**
> >
> > [A] Dequan Wang, Evan Shelhamer, Shaoteng Liu, Bruno Olshausen, and Trevor Darrell. Tent: Fully test-time adaptation by entropy minimization. In ICLR, 2021.
> >
> > [B] Shuaicheng Niu, Jiaxiang Wu, Yifan Zhang, Zhiquan Wen, Yaofo Chen, Peilin Zhao, and Mingkui Tan. Towards stable test-time adaptation in dynamic wild world. In ICLR, 2023.
> >
> > [C] Jielin Qiu, Yi Zhu, Xingjian Shi, Florian Wenzel, Zhiqiang Tang, Ding Zhao, Bo Li, and Mu Li. *Benchmarking Robustness of Multimodal Image-Text Models under Distribution Shift*. Journal of Data-centric Machine Learning Research, 2023.
> >
> > [D] Xiaoqiang Lu, Binqiang Wang, Xiangtao Zheng, and Xuelong Li. *Exploring models and data for remote sensing image caption generation*. IEEE Transactions on Geoscience and Remote Sensing, 2017.
> >
> > [E] Zhiqiang Yuan, Wenkai Zhang, Kun Fu, Xuan Li, Chubo Deng, Hongqi Wang, and Xian Sun. *Exploring a fine-grained multiscale method for cross-modal remote sensing image retrieval*. IEEE Transactions on Geoscience and Remote Sensing, 2021.
> >
> > [F] Shuaicheng Niu, Jiaxiang Wu, Yifan Zhang, Yaofo Chen, Shijian Zheng, Peilin Zhao, and Mingkui Tan. Efﬁcient test-time model adaptation without forgetting. In ICML, 2022.
> >
> > [G] Jonghyun Lee, Dahuin Jung, Saehyung Lee, Junsung Park, Juhyeon Shin, Uiwon Hwang, and Sungroh Yoon. Entropy is not enough for test-time adaptation. In ICLR, 2024.

---

> > > ### Author Response · Authors · 2024-11-24
> > >
> > > Dear reviewer Wks8,
> > >
> > > We would like to know if our response has addressed your concerns and questions. If you have any further concerns or suggestions for the paper or our rebuttal, please let us know. We would be happy to engage in further discussion and manuscript improvement.
> > >
> > > Thank you again for the time and effort you dedicated to reviewing this work.

---

### Meta-Review · Area_Chair_PRz7 · 2024-12-17

**Metareview:**

This paper tackles the challenging 'query shift' problem in cross-modal retrieval, in which the distribution of query data deviates from the source domain. They formulate their algorithm as a test-time adaptation for cross-modal retrieval, which includes a query prediction refinement module to refine the query predictions and a joint objective function to prevent the disturbances caused by the query shift. Sufficient experimental results on multiple datasets show the effectiveness of the proposed method in such different cases.

In the first round, the main concerns lie in the types of query shifts, robustness to a diverse set of query shifts, parameters, and other minor issues. After rebuttal, all these concerns were eliminated, and all the reviewers voted to accept this work.

After carefully reading the main document and supplementary docs, the AC has the following comments about this paper:
***Advantages***
This paper addresses the query shift problem in cross-modal retrieval during test time. The problem itself is practical due to the diverse and unpredictable nature of open-world queries. Through extensive experimental analyses, the authors identify two key challenges posed by query shift and propose an effective method to address them. The manuscript and appendix include thorough empirical studies validating the significance of the addressed problem and the effectiveness of the proposed approach.
***Future Direction***
Although the authors pointed out the query shift issue in cross-modal retrieval, how to effectively handle more potential scenarios contaminated with query shift should be further discussed and highlighted.

In summary, due to novel problem definition, technical context and experimental results, this paper is a good work to be presented in ICLR 2025.

**Additional Comments On Reviewer Discussion:**

All four reviewers recommend acceptance, with three giving clear acceptance and one providing a borderline acceptance. The reviewers recognize the contributions of this work. Overall, I believe this paper provides valuable insights for the test-time adaptation (TTA) and cross-modal retrieval communities and strongly recommend its acceptance.

---

### Decision · Program_Chairs · 2025-01-22

Accept (Spotlight)